# Finite Sample Bounds for Non-Parametric Regression: Optimal Sample Efficiency and Space Complexity

**Davide Maran**                                                                *davide.maran@polimi.it*
*Department of Electronics, Information and Bioengineering*
*Politecnico di Milano*
**Marcello Restelli**                                                        *marcello.restelli@polimi.it*
*Department of Electronics, Information and Bioengineering*
*Politecnico di Milano*

Reviewed on OpenReview: *https://openreview.net/forum?id=7AHO204EaZ*

## Abstract

We address the problem of learning an unknown smooth function and its derivatives from noisy pointwise evaluations under the supremum norm. While classical nonparametric regression provides a strong theoretical foundation, traditional kernel-based estimators often incur high computational costs and memory requirements that scale with the sample size, limiting their utility in real-time applications such as reinforcement learning. To overcome these challenges, we propose a parametric approach based on a finite-dimensional representation that achieves minimax-optimal uniform convergence rates. Our method enables lightweight inference without storing all samples in memory. We provide sharp finite-sample bounds under sub-Gaussian noise, derive second-order Bernstein-type guarantees, and prove matching lower bounds, thereby confirming the optimality of our approach in both estimation error and memory efficiency.

## 1 Introduction and Motivation

Regression is a fundamental task in machine learning and statistics, and it is perhaps the most classical family of problems to be studied in these fields. While traditional regression techniques, such as linear models, are well-understood and widely used, many real-world applications require learning smooth underlying functions from noisy data. This challenge is omnipresent in Science, particularly in fields such as signal processing, bioinformatics, and financial modeling, where extracting meaningful patterns from noisy observations is essential (Bishop & Nasrabadi, 2006; LeCun et al., 2015). In this paper, we focus on the problem of regression for a function $f : [-1, 1]^d \to \mathbb{R}$, which is known to be smooth, i.e., differentiable a given number of times, and aim to obtain an estimate that is uniformly accurate over the entire domain. This kind of guarantee, which is not dataset-related, requires the algorithm to also specify the values $\{x_i\}_{i=1}^n$ for which evaluations of $f$ are needed, a procedure known as *passive design*. Non-parametric methods, such as kernel regression, Gaussian Processes, and local polynomial estimators, are popular choices for this task due to their flexibility and ability to capture complex data patterns without assuming a specific functional form (Williams & Rasmussen, 2006; Hastie et al., 2009). However, this flexibility often comes at the cost of computational efficiency and scalability. Non-parametric models typically require storing and processing the entire dataset during inference, making them less practical for large-scale applications or real-time systems (Wasserman, 2006).

The limitations outlined above significantly restrict the applicability of classical nonparametric statistical theory to the modern machine learning settings where it would be most valuable. In particular, uniform approximation guarantees are highly desirable in contemporary reinforcement learning and bandit problems, where worst-case control over continuous state or action spaces is essential for stability, policy evaluation, and reliable optimization. In that contest, interest in Sobolev-like assumptions has grown significantly in recent years (Liu et al., 2021; Maran et al., 2024c;a).The theoretical frameworks developed in these areas typically use parametric predictors and require finite-sample guarantees under sub-Gaussian noise, reflecting the statistical requirements of sequential decision-making. This stands in contrast with much of the nonparametric regression literature, which prioritizes asymptotic optimality and relies on non-parametric methods.

Such discrepancies underscore the need for a paradigm that bridges nonparametric guarantees with parametric efficiency. Parametric models offer superior scalability and computational efficiency, with their memory footprint independent of dataset size (Murphy, 2012), as well as several other advantages, including interpretability and compactness (Bishop & Nasrabadi, 2006). Unfortunately, while parametric methods are naturally suited for $L^2$ (mean-squared-error) guarantees, they often struggle to provide the uniform error control over the entire domain required by modern machine learning.

### 1.1 Contributions and paper structure

In this paper, we propose the first parametric algorithm that achieves minimax-optimal sample complexity for nonparametric regression under sub-Gaussian noise and passive (non-adaptive) design, via a fully finite-sample analysis that yields results holding in high probability. Precisely, our contributions are the following:

1. **Minimax-optimal uniform estimation**. We introduce a parametric estimator that achieves optimal finite-sample rates in the supremum norm for the regression of smooth functions, matching the classical minimax rates of nonparametric regression. Our guarantees hold uniformly over the entire domain and extend to the simultaneous estimation of all derivatives up to the smoothness order: the estimator for $f^{(\alpha)}(\cdot)$ is the $\alpha-$derivative of $\widehat{f}_n(\cdot)$. This property, often referred to as the *plug-in* estimation (Bickel & Ritov, 2003), is particularly valuable in modern ML (Liu & Li, 2023) since it eliminates any trade-off in the choice of hyperparameters.

2. **Finite-sample analysis and second order bounds**. We provide high-probability finite-sample bounds under sub-Gaussian noise, making the bias–variance trade-off explicit and avoiding reliance on purely asymptotic arguments. The resulting rates recover the classical optimal scaling while remaining valid for any finite number of samples, with an explicit and optimal dependence on all problem parameters. Beyond standard sub-Gaussian analyses, we derive *Bernstein-type* error bounds that exploit variance information, yielding sharper guarantees when the noise has a large range and small variance. To the best of our knowledge, this is the first second-order finite-sample analysis for uniform estimation in this setting. Second-order bounds have proved to be the backbone of tight sample-complexity guarantees for modern ML in several scenarios (Cesa-Bianchi et al., 2007; Wang et al., 2024; Pacchiano, 2024; Olkhovskaya et al., 2023).

3. **Computational and memory complexity**. Unlike classical nonparametric methods, our approach yields a lightweight predictor whose memory and computational costs at prediction time depend only on the number of parameters. We show that this complexity is information-theoretically optimal by proving a matching lower bound on the memory required by any statistically optimal estimator.

4. **Numerical validation.** Empirical evaluations on real-world datasets, augmented with synthetic noise, demonstrate that our method achieves error rates comparable to the state of the art while requiring only a fraction of the computational overhead, thereby validating our theoretical findings.

All the results in the paper are proved under periodic boundary conditions, as is often done when using a Fourier feature map Liu & Li (2023). In the appendix G, we show how to generalize them to non-periodic functions.

**Paper structure** The rest of the paper is organized as follows: after introducing the necessary notation in Section 2, we establish the foundation of our approach based on the properties of the Fourier Series (see Section 3). The algorithm is derived in Section 4, along with its sample complexity guarantee, which is first presented for sub-Gaussian noise and then as a second-order bound. A finite-time lower bound is presented in Section 5, which matches the previous upper bound in all problem-dependent constants. As anticipated, the only algorithms that achieve optimal sample complexity are nonparametric; we introduce them in Section 6 and compare them with DUPA with respect to computational/space complexity, as well as empirical performance (see Appendix 7).

## 2 Preliminaries

In this paper, we focus on a **regression** problem, i.e., approximating a black-box function $f : \mathcal{X} \subset \mathbb{R}^d \to [0, 1]$ from noisy samples $y_i$ corresponding to some $x_i \in \mathcal{X}$.

**Assumption 1.** *(Passive design with sub-Gaussian noise) The agent is able to choose $n$ query points in advance, receiving a dataset of samples $\{x_i, y_i\}_{i=1}^n$ given by realizations of $Y_i$ such that $\mathbb{E}[Y_i|x_i] = f(x_i)$ and $Y_i - f(x_i)$ is $\sigma$-sub-Gaussian independent from $Y_j$ for $j \neq i$.*

We study the case where the function $f$ is smooth. Given a multi-index $\boldsymbol{\alpha}$ as a tuple of non-negative integers $(\alpha_1, \dots \alpha_d)$, we say that $f \in \mathcal{C}^\nu(\mathcal{X})$, for $\nu \in (0, +\infty)$, if it is $\nu_*$−times continuously differentiable for $\nu_* := \lceil \nu - 1 \rceil$, and there exists a constant $L_\nu(f)$ such that:

$$\forall \boldsymbol{\alpha} : \|\boldsymbol{\alpha}\|_1 = \nu_*, \qquad \forall x, y \in \mathcal{X} \left| D^{(\boldsymbol{\alpha})} f(x) - D^{(\boldsymbol{\alpha})} f(y) \right| \leq L_\nu(f) \|x - y\|_\infty^{\nu - \nu_*}. \tag{1}$$

The multi-index derivative is defined as $D^{(\boldsymbol{\alpha})} f := \frac{\partial^{\alpha_1 + \dots + \alpha_d}}{\partial x_1^{\alpha_1} \dots \partial x_d^{\alpha_d}}$. The previous set becomes a normed space when endowed with a norm $\|f\|_{\mathcal{C}^\nu}$ defined as $\max \left\{ \max_{|\boldsymbol{\alpha}| \leq \nu_*} \|D^{(\boldsymbol{\alpha})} f\|_{L^\infty}, L_\nu(f) \right\}$. Note that, when $\nu \in \mathbb{N}$, this norm reduces to $\|f\|_{\mathcal{C}^\nu} = \max_{|\boldsymbol{\alpha}| \leq \nu} \|D^{(\boldsymbol{\alpha})} f\|_{L^\infty}$, since the Lipschitz constant $L_\nu(f)$ of the derivatives up to order $\nu_* = \nu - 1$ corresponds exactly to the upper bound of the derivatives of order $\nu$ (which exists as a Lipschitz function is differentiable almost everywhere).

**Assumption 2.** *(Smooth and periodic function) $f(\cdot) \in \mathcal{C}_p^\nu([-1, 1]^d)$ for some known $\nu > 0$ and an upper bound for $\|f\|_{\mathcal{C}^\nu}$ is known.*

Assumption 2 contains, in addition to smoothness, the requirement of periodicity at the boundaries of $[-1, 1]^d$. While this assumption may seem restrictive, we argue that it does not make the problem any easier. In fact, we will prove a result showing that the same lower bound for the general case holds for the case with periodicity (theorem 10). Moreover, under the assumption that we can query samples on a larger set that contains $\mathcal{X}$, we shall prove that our algorithm can be adapted to a general non-periodic function (see Appendix G). The method introduced in this paper is based on the theory of the Fourier Series. This discipline focuses on spaces of functions that are endowed with periodic boundary conditions on the interval. In the following section, to keep things intuitive, we will only focus on the one-dimensional case, while the statistical complexity bound for arbitrary dimension $d$ will be given in section 4.1.

**Periodic functions and Fourier series** For now, without loss of generality, we keep $\mathcal{X} = [-1, 1]$. We define trigonometric polynomial of degree $N$ as a function $f$ that can be written as $f(x) = a_0 + \sum_{t=1}^N b_t \sin(t\pi x) + a_t \cos(t\pi x)$, for real coefficients $\{a_t, b_t\}$. Obviously, any function of this kind is periodic on $[-1, 1]$. By convenience, we can stack the coefficients of this representation into a unique vector $\theta$, instead of having $a, b$. Any trigonometric polynomial can be written as

$$f(x) = \sum_{t=-N}^N \theta_t \operatorname{soc}(t, x) =: \phi_N(x)^\top \theta, \tag{2}$$

where $\phi_N(x)$ is the Fourier feature map, which contains terms of the form $\sin(t\pi x)$ or $\cos(t\pi x)$ for $t \leq N$. In the following, we are going to use the definition corresponding to Equation (2) and call the vector space of functions defined in this way as $\mathbb{T}_N$. Periodic functions, and trigonometric polynomials in particular, admit a convolution operator that will be useful in the following analyses. In the rest of this paper, we will call, for periodic functions $f, g$ on $[-1, 1]$, $f * g(x) := \int_{-1}^1 f(y) g(x - y) \, dy$, where $g$ is extended by periodicity: for $z > 1$, $g(z) := g(z - 2)$ and for $z < -1$, $g(z) := g(z + 2)$. This operator satisfies the usual properties of convolution and is usually called *circular convolution*.

## 2.1 Regression and Fourier Series

The underlying idea of the Fourier Series is to use trigonometric polynomials to approximate generic functions. In fact, any function $f(\cdot)$ can be written as a trigonometric polynomial plus an error term whose magnitude can be bounded, for example, with the following classical result.

**Theorem 3.** *(Theorem 4.1 part (ii) from Schultz (1969)) There exists an absolute constant $K > 0$ such that, for any $f \in C_p^\nu([-1, 1]^d)$ we have:*

$$\inf_{T_N \in \mathbb{T}_N} \|T_N(\cdot) - f(\cdot)\|_{L^\infty} \leq K N^{-\nu} \|f\|_{\mathcal{C}^\nu}.$$

Note that the trigonometric polynomial realizing the infimum of the previous definition exists but does *not* correspond to the Fourier projection on $\mathbb{T}_N$, which instead is defined by minimizing the $L^2$−norm. For the latter notion of distance,

results analogous to the one presented in theorem 3 are known, which always link the smoothness of a function to the approximation rate.

In the context of Learning theory (Vapnik, 2013), and specifically Regression (Harrell et al., 2001), these kinds of results have inspired a simple yet effective idea. If the target function $f(\cdot)$ is smooth, and we can access it through noisy samples, we can choose $N$ so that the error term is negligible and then run the simple Linear Regression algorithm (Montgomery et al., 2021) to estimate coefficients $\widehat{\theta}_n$ such that $\phi_N(\cdot)^\top \widehat{\theta}_n \approx f(\cdot)$. In order to get a satisfactory result, we need two things to be high: 1) $N$, so that the approximation error becomes negligible (low bias); 2) $n$, the number of samples, so that the variance is also low. In fact, the two needs cannot be disentangled: the higher $N$, the more samples are needed to avoid overfitting.

### 2.1.1 The problem of linear regression under misspecification

Formalizing our setting, we have to fit a function $f(\cdot) = \phi_N^\top(\cdot)\theta + \xi_N[f](\cdot)$ from noisy observation, knowing that $\|\xi_N[f](\cdot)\|_{L^\infty} \leq \xi_N^\infty \to 0$ with $N \to \infty$, at a certain known rate. In case we are interested in minimizing the $L^2$ loss, things go smooth, as from it follows that, sampling uniformly on $[-1, 1]$ we can guarantee Wainwright (2019) (Chapters 13 and 14) $\|f(\cdot) - \langle \phi_N(\cdot), \widehat{\theta}_n \rangle\|_{L^2}$

$$\propto \sqrt{\mathop{\mathbb{E}}_{X \sim \mathrm{Unif}(-1,1)}[(f(X) - \langle \phi_N(X), \widehat{\theta}_n \rangle)^2 | \widehat{\theta}_n]} = \mathcal{O}\left(\sqrt{N/n} + \xi_N^\infty\right).$$

In light of this result, optimal error bounds for Fourier regression in the case of the $L^2$ were proved Tsybakov (2009) (1.7 Projection estimators). The pain starts when focusing on the uniform error instead. In fact, Lattimore et al. (2020) proved that for general feature maps of length $N$, the best provable guarantee takes the following form

$$\|f(\cdot) - \langle \phi_N(\cdot), \widehat{\theta}_n \rangle\|_{L^\infty} = \mathcal{O}\left(\sqrt{N/n} + \sqrt{N}\xi_N^\infty\right).$$

This dependence on $\sqrt{N}$, coupled with the misspecification, is very unfortunate in our case. Not only can it be shown to be suboptimal, but it may also lead to vacuous statistical complexity bounds (Maran et al., 2024c). As we anticipated, the whole idea of using the Fourier Series in Regression builds on the fact that the approximation term $\xi_N^\infty$ vanishes for $N \to \infty$, a property that may not hold for the product $\sqrt{N}\xi_N^\infty$. Two more things are particularly discouraging: 1) the result was proved for the *bandit* setting, where the learner can adaptively choose the queries $x_i$ based on past observations $y_i$, which is easier than our passive design assumption; 2) we are interested in a uniform bound also for the derivatives of the function $f$, and the misspecification is known to be amplified for this kind of estimation problems. To elude this result, we need some argument that does not come from linear algebra - as anything that holds for generic feature maps is doomed by the previous negative result - but something that exploits the specific features of the Fourier basis.

## 3 Approximation via Convolution Kernels

The most standard way to find the Fourier coefficients of a given function $f \in L^2([-1, 1])$ is by performing a scalar product with the base functions $\sin(t\pi\cdot)$ and $\cos(t\pi\cdot)$. This procedure gives the value of each of the coefficients of the trigonometric polynomial in equation 2. Still, there is another way to perform the same operation, which directly gives the Fourier series truncated at a given order $N$; this passes through a function called *Dirichlet Kernel*, defined as

$$D_N(x) := 1 + 2\sum_{t=1}^{N} \cos(t\pi x) = \frac{\sin((N + 1/2)\pi x)}{\sin(\pi x/2)}. \tag{3}$$

In fact, calling $\mathcal{F}_N[f](\cdot)$ the Fourier Series of $f$ truncated to the term $N$, it was proved that $\mathcal{F}_N[f](\cdot) = D_N * f(\cdot)$. As we shall see, expressing this operator in terms of convolution is fundamental to what follows. However, first note that finding the Fourier series is not exactly what we are looking for. In fact, as we want a method that achieves a performance in infinity norm, we are interested in finding the trigonometric polynomial $T_N \in \mathbb{T}_N$ minimizing $\|T_N(\cdot) - f(\cdot)\|_{L^\infty}$. Instead, by using Fourier Series, we minimize $\|T_N(\cdot) - f(\cdot)\|_{L^2}$. Unfortunately, the difference

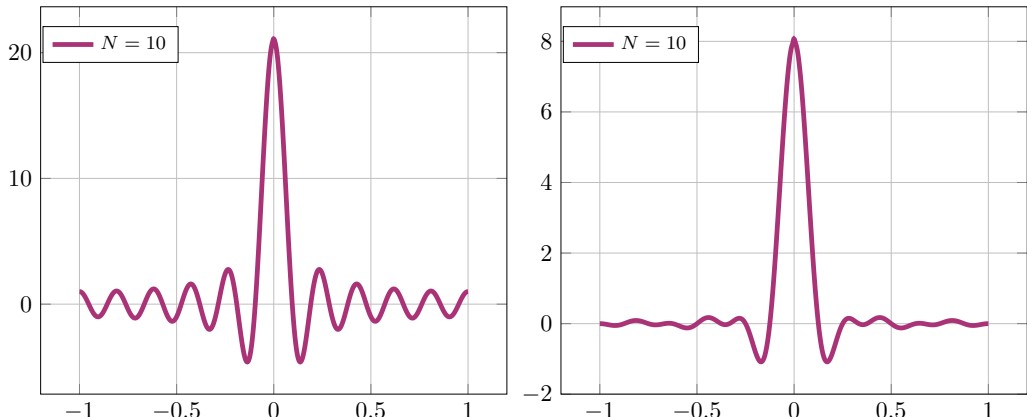

Figure 1: Dirichlet kernel (left, (3)) and De la Vallée-Poussin kernel (right, (4)).The degree $N$ denotes the number of terms retained in the Fourier basis expansion. As $N \to \infty$, both kernels converge weakly to the Dirac delta function. Crucially, the Dirichlet kernel exhibits pronounced oscillations (Gibbs phenomenon), whereas the de la Vallée Poussin kernel is well-localized, remaining significantly greater than zero only in a narrow neighborhood around the origin. Image adapted from (Maran et al., 2024b).

between these two objectives is substantial: we cannot replace one with the other without significantly weakening the guarantees. Luckily, a solution can be found by replacing $D_N$ with another kernel, defined as

$$V_N(x) := \frac{1}{N/2+1} \sum_{t=N/2}^{N} D_t(x). \tag{4}$$

This function, known as the De la Valée Poussin kernel (de la Vallée Poussin, 1918; De La Vallée Poussin et al., 1919), possesses important properties. Even if, contrary to its $L^2$ counterpart, it is not able to provide with $V_N * f(\cdot)$ the exact trigonometric polynomial in $\mathbb{T}_N$ which minimizes the $L^\infty$ error, it comes close. In fact, the result of the convolution operator is just a constant time worse than the optimal projection in $\|\cdot\|_{L^\infty}$, as the next theorem states.

**Theorem 4.** *Let $f \in \mathcal{C}_p^\nu([-1,1])$ and $\alpha \in \{0, 1, \dots \nu_*\}$. Then, $V_N * f(\cdot) \in \mathbb{T}_N$, and we have for a constant $K > 0$,*

$$\|f^{(\alpha)}(\cdot) - (V_N * f)^{(\alpha)}(\cdot)\|_{L^\infty} \le K(2/N)^{\nu-\alpha}\|f\|_{\mathcal{C}^\nu}.$$

Comparing this result to the one of theorem 3, we can see that the convolution $V_N * f$ not only approximates the function $f$ but also all its derivatives, each one with optimal order. The proof, which is left to appendix B, is based on a simple property, which is also going to be useful later,

$$\|V_N(\cdot)\|_{L^1} \le \Lambda_1, \tag{5}$$

for an absolute constant $\Lambda_1$, that, crucially, does not depend on $N$, and has recently proved to be $\frac{1}{3} + \frac{2\sqrt{3}}{\pi} \approx 1.43$ (Mehta, 2015). Thanks to this equation, we can split $V_N$ into a positive and a negative part, both with finite integrals. We will call these two parts $V_N^+(\cdot), V_N^-(\cdot) \ge 0$ as

$$V_N(\cdot) =: \beta_+ V_N^+(\cdot) - \beta_- V_N^-(\cdot) \qquad \int_{-1}^{1} V_N^+(x)\,dx = \int_{-1}^{1} V_N^-(x)\,dx = 1. \tag{6}$$

## 3.1 The Perturbation Trick

For now, we have seen that the abstract idea of projecting, w.r.t. the infinity norm, a function onto the vector space $\mathbb{T}_N$ can be performed through a convolution with $V_N(\cdot)$. A key advantage of this approach is that we can approximate all derivatives of a function at the same time: taking the derivative of the convolution of a function $f(\cdot)$ with kernel

---

**Algorithm 1** DUPA: Derivative-Uniform Parametric Approximation

---

**Require:** Query space $[-1, 1]$, max number of samples $n$, feature map length $N$, smoothness order $\nu > 0$
**Ensure:** Estimated function $\widehat{f}_n$ and its derivatives $\widehat{f}_n^{(1)}, \ldots, \widehat{f}_n^{(\nu_\star)}$
1: $\varepsilon \leftarrow 1/(8\pi N)$
2: Find $\varepsilon$-cover $\mathcal{C}_\varepsilon$ of $[-1, 1]$
3: Let $\mathcal{X}_\phi := \{\phi_N(x) : x \in \mathcal{C}_\varepsilon\}$
4: Compute quasi-optimal design $\rho$ for the set $\mathcal{X}_\phi$
5: Compute $V_N^+, V_N^-, \beta_+, \beta_-$ from Equation (6)
6: $n_{\text{tot}} \leftarrow \lfloor n/4 \rfloor$
7: $i \leftarrow 0$
8: **for all** $x \in \text{supp}(\rho)$ **do**
9:     **for** $j = 1$ to $\lceil n_{\text{tot}} \cdot \rho(x) \rceil$ **do**
10:         $x_i \leftarrow x$
11:         Sample $\eta_+ \sim V_N^+(\cdot)$
12:         Sample $\eta_- \sim V_N^-(\cdot)$
13:         Query $y_i^+$ at $x + \eta_+$
14:         Query $y_i^-$ at $x + \eta_-$
15:         $y_i \leftarrow \beta_+ y_i^+ - \beta_- y_i^-$
16:         $i \leftarrow i + 1$
17:     **end for**
18: **end for**
19: Solve least squares: $\widehat{\theta}_n = \arg\min_{\theta \in \mathbb{R}^N} \sum_i (\phi_N(x_i)^\top \theta - y_i)^2$
20: **return** $\widehat{f}_n(\cdot) = \phi_N(\cdot)^\top \widehat{\theta}_n$ and its derivatives

---

$V_N(\cdot)$ is the same as making the convolution of the derivative of $f(\cdot)$ with $V_N(\cdot)$. Therefore, by approximating the function itself, we implicitly approximate all of its derivatives. However, the real advantage of this approach lies in the possibility to choose the sampling points (even before knowing any of the labels). As we have seen in section 2.1.1, problems arise when performing linear regression w.r.t. the basis $\phi_N(\cdot)$ of a function that is not written exactly as $\phi_N(\cdot)^\top \theta$. While our target function $f(\cdot)$ cannot be written in this form, this holds for $V_N * f(\cdot)$, as Theorem 4 ensures. Here is the crucial point: while we are interested in approximating $f(\cdot)$, gathering samples from $V_N * f(\cdot)$ is better, as it approximates $f(\cdot)$ at the optimal order and is perfectly linear in $\phi_N(\cdot)$. Unfortunately, the agent-environment interaction does not allow sampling from the latter function. For this reason, one needs to rely on a randomization trick.

Recall that, when perturbing the argument of a function $g(\cdot)$ with a noise $\eta$ of density function $f_\eta(\cdot)$, we have $\mathbb{E}[g(\cdot + \eta)] = g * f_\eta(\cdot)$. If we apply the previous principle with a noise of density $\eta \sim V_N(\cdot)$, we get $\mathbb{E}[g(\cdot + \eta)] = g * V_N(\cdot)$. Unfortunately, $V_N(\cdot)$ is *not* a density: it is not positive. But luckily, using the decomposition (6) does the job:

$$\eta_+ \sim V_N^+(\cdot), \ \eta_- \sim V_N^-(\cdot) \qquad \beta_+ \mathbb{E}[g(\cdot + \eta_+)] - \beta_- \mathbb{E}[g(\cdot + \eta_-)] = g * V_N(\cdot). \tag{7}$$

The last equation 7 is what we are going to use to fool our own learner. The idea of our algorithm is to start from a linear learner which queries data $\bar{x}_1 \ldots \bar{x}_{n'}$ from a distribution $\rho$. Instead of querying the model for the point chosen by the learner, we ask for points $x_1 \ldots x_{2n'}$ by perturbing the original points as in equation 7. After receiving the outputs $y_1 \ldots y_{n'}$, the linear learner will act as if the target function were $V_N * f(\cdot)$, which is linear without any misspecification.

In the next section, we formalize this idea as Algorithm 1 and prove its theoretical guarantees. This trick, known as *projection by convolution*, was recently used by Maran et al. (2024b) in reinforcement learning.

## 4 The DUPA Algorithm: Design and Bounds

Algorithm 1, DUPA, takes as input all the problem parameters plus the order $N$ of the feature map we are going to use, even if, as we shall see, specific values of $N$ are required to get theoretical guarantees. The first lines are standard; we start finding an $\varepsilon$ cover of the interval $[-1, 1]$ and applying the feature map to each point. In line 4, the linear learner chooses which points they desire to query the black-box model. In this step, we employ the notion of

quasi-optimal design for the least square problem, which we explained in detail in appendix C. Keeping things simple, using a quasi-optimal design means finding the best distribution of data to fit linear regression if we are interested in the supremum error on a finite set of points, $\mathcal{C}_\varepsilon$ in our case. Using a quasi-optimal design reduces the number of queries by a factor of $\sqrt{N}$ while incurring only $\log(|\mathcal{C}_\varepsilon|)$ queries, and is necessary to achieve the optimal sample complexity. The noise densities follow exactly equation (6), and the sampling process, at lines 13,14 follows exactly equation (7). After these steps, the algorithm proceeds exactly as a simple linear regression. Thanks to this trick, algorithm 1 achieves an optimal sample complexity guarantee for all the derivatives of the target function $f$, as the next theorem states.

**Theorem 5.** *Run* DUPA *Algorithm 1 for a choice $N$ such that $16N \log \log(N) < n$. Under Assumptions 1,2 for $d = 1$, with probability at least $1 - \delta$, the output satisfies, for all $\alpha = \{0, \ldots \nu_*\}$*

$$\|f^{(\alpha)}(\cdot) - \phi_N^{(\alpha)}(\cdot)^\top \widehat{\theta}_n\|_{L^\infty} \lesssim N^\alpha \left( \frac{\|f\|_{\mathcal{C}^\nu}}{N^\nu} + \sqrt{\frac{N \log(n/\delta)}{n}} \sigma \right), \tag{8}$$

*for $\|f\|_{\mathcal{C}^\nu}, \sigma \geq 1$.*

**Proof Sketch 1.** *The proof begins by formalizing the idea of sampling $\eta_+, \eta_-$ that we heuristically introduced in Section 3.1. This makes the Lebesgue constant of the DVP kernel emerge, which is bounded. Then, the total error is partitioned into bias and variance terms, which are treated separately. The demonstration is then reduced to three parts, which encompass the key steps of the theory developed so far. The first is to ensure that the discretization over the $\mathcal{C}_\varepsilon$ points causes no harm. This is not trivial due to the requirement of uniform approximation, but can be solved thanks to the particular properties of trigonometric polynomials. The second is to use optimal design theory to bound the error due to sample stochasticity. The last one bounds the bias of approximation by a trigonometric polynomial, generalizes Theorem 3, and puts everything together.*

The former result is useful for explicitly showing the bias-variance trade-off. The bias term decays as $N^{\alpha - \nu}$, the typical rate granted by Jackson's theorems. The variance decays as $\sqrt{N \log(N/\delta)/n}$, which is the standard rate for linear regression with an optimal design. The bound is completed by explicitly choosing the feature length $N$.

**Corollary 6.** *Run* DUPA *algorithm 1 for $N = \left( \frac{n}{\log(n/\delta)} \right)^{\frac{1}{2\nu+1}} \|f\|_{\mathcal{C}^\nu}^{\frac{2}{2\nu+1}} \sigma^{-\frac{2}{2\nu+1}}$. Under assumptions 1,2 for $d = 1$, we have with probability at least $1 - \delta$, for all $\alpha \in \{0, \ldots \nu_*\}$*

$$\|f^{(\alpha)}(\cdot) - \phi_N^{(\alpha)}(\cdot)^\top \widehat{\theta}_n\|_{L^\infty} \lesssim \left( \frac{n}{\log(n/\delta)} \right)^{-\frac{\nu-\alpha}{2\nu+1}} \|f\|_{\mathcal{C}^\nu}^{\frac{2\alpha+1}{2\nu+1}} \sigma^{\frac{2\nu-2\alpha}{2\nu+1}}, \tag{9}$$

*for $\|f\|_{\mathcal{C}^\nu}, \sigma \geq 1$.*

The proof follows by substituting the given value of $N$ and noting that $\log(N) \asymp \log(n/\log n) \asymp \log(n)$. This rate matches the asymptotic rate of Stone (1982), which is optimal for nonparametric regression.

**The importance of selecting the De la Vallée Poussin kernel**  In the previous section, we emphasized the critical importance of employing the De la Vallée Poussin kernel rather than the simpler and more intuitive Dirichlet kernel. Indeed, DUPA remains functional in principle when implemented with the Dirichlet kernel; one would only need to modify line 13 and the subsequent one. However, such an implementation fails to achieve optimal sample complexity. By derivation, it can be shown that the resulting upper bound incurs a multiplicative factor determined by the Lebesgue constant (i.e., the $L^1$ norm of the kernel), which for the Dirichlet kernel scales as $\log(N)$. This factor multiplies the $\log(n)$ term in the upper bound, thereby compromising its optimality.

## 4.1  Multi-dimensional generalization

After showing that our algorithm works for the one-dimensional setting, we generalize the result to the regression problem of a function $f : [-1, 1]^d \to \mathbb{R}$. The Fourier theory for multivariate functions is similar to that of the univariate case (Katznelson, 2004), and approximation theorems hold in the same way. Just, the feature map $\phi_N(\cdot)$, whose length was $2N + 1$, gets replaced by $\boldsymbol{\phi}_N(\cdot)$, which contains interaction terms among the $d$ variables, and for this reason has length $N_d \approx N^d$. This worsens the results, as expected due to the infamous curse of dimensionality. Nonetheless, our algorithm 1 is substantially unchanged, just we replace $\phi_N(\cdot)$ with $\boldsymbol{\phi}_N(\cdot)$ and $V_N(\cdot)$ with the multidimensional Vallèe de la Poussin's kernel (Németh, 2016).

| Algorithm | Time training | Time prediction | Space training | Space prediction |
|---|---|---|---|---|
| LPE | $\mathcal{O}(n)$ | $\mathcal{O}(nm)$ | $\mathcal{O}(n)$ | $\mathcal{O}(n)$ |
| KERNEL RIDGE REG. | $\mathcal{O}(n^3)$ | $\mathcal{O}(nm)$ | $\mathcal{O}(n^2)$ | $\mathcal{O}(nm)$ |
| DUPA (algorithm 1) | $\mathcal{O}\left(n^{\frac{2\nu+3d}{2\nu+d}}\right)$ | $\mathcal{O}\left(mn^{\frac{d}{2\nu+d}}\right)$ | $\mathcal{O}\left(n^{\frac{2d}{2\nu+d}}\right)$ | $\mathcal{O}\left(n^{\frac{d}{2\nu+d}}\right)$ |

Table 1: Table containing the computational complexities of the algorithms with optimal statistical efficiency. Number of training samples: $n$, prediction samples: $m$.

**Theorem 7.** *Run* DUPA *algorithm 1 for*

$$N = \left(\frac{n}{\log(n/\delta)}\right)^{\frac{1}{2\nu+d}} \|f\|_{\mathcal{C}^\nu}^{\frac{2}{2\nu+d}} \sigma^{-\frac{2}{2\nu+d}}.$$

*Under assumptions 1,2 we have with probability at least $1-\delta$, for all $|\boldsymbol{\alpha}| \leq \nu_*$,*

$$\|D^{(\boldsymbol{\alpha})}f(\cdot) - D^{(\boldsymbol{\alpha})}\boldsymbol{\phi}_N(\cdot)^\top \widehat{\boldsymbol{\theta}}_n\|_{L^\infty} \lesssim \left(\frac{n}{\log(n/\delta)}\right)^{-\frac{\nu-|\boldsymbol{\alpha}|}{2\nu+d}} \|f\|_{\mathcal{C}^\nu}^{\frac{2|\boldsymbol{\alpha}|+d}{2\nu+d}} \sigma^{\frac{2\nu-2|\boldsymbol{\alpha}|}{2\nu+d}}, \tag{10}$$

*for $\|f\|_{\mathcal{C}^\nu}, \sigma \geq 1$.*

Once again, the rate in $n$ was proved to be asymptotically optimal for nonparametric regression.

## 4.2 Second-order bound

In the previous section, we demonstrated that the infinite-norm error can be bounded with high probability under the assumption of sub-Gaussian noise. This choice, although standard in the modern statistical complexity literature due to its analytical tractability, can be overly conservative in scenarios where the local dispersion of the noise is significantly smaller than its global range. To overcome this limitation, it is possible to resort to a Bernstein-type second-order bound, which allows for refinement of the tail control of the distribution by integrating variance information.

In this section, we replace assumption 1 with the following.

**Assumption 8.** *(Passive design with bounded noise) The agent is able to choose $n$ query points in advance, receiving a dataset of samples $\{x_i, y_i\}_{i=1}^n$ given by realizations of $Y_i$ such that $\mathbb{E}[Y_i] = \mathbb{E}[f(x_i)]$ and $Y_i - f(x_i)$ is independent from $Y_j$ for $j \neq i$ and satisfies*

$$Var[Y_i - f(x_i)] \leq \gamma^2 \qquad |Y_i - f(x_i)| \leq B \text{ a.s.}$$

Apart from introducing $\gamma$, the former assumes that the noise is bounded in $[-B, B]$. Any bounded random variable is sub-Gaussian, and the following relation holds: $\gamma \leq \sigma \leq B$. Even if not all sub-Gaussian random variables are bounded, the following equation holds for a sequence of this type

$$\mathbb{P}\left(\max_{t=1,\ldots n} |\xi_t| \leq \sigma\sqrt{\log(n/\delta)}\right) \geq 1 - \delta.$$

Therefore, any sample complexity bound can be easily generalized from the bounded to the sub-Gaussian case. The next theorem shows a second-order bound for the sample complexity of nonparametric regression.

**Theorem 9.** *Run* DUPA *algorithm 1 for $N = \left(\frac{n}{\log(n/\delta)}\right)^{\frac{1}{2\nu+d}} \|f\|_{\mathcal{C}^\nu}^{\frac{2}{2\nu+d}}$. Under assumptions 8,2 we have with probability at least $1-\delta$, for all $|\boldsymbol{\alpha}| \leq \nu_*$,*

$$\|D^{(\boldsymbol{\alpha})}f(\cdot) - D^{(\boldsymbol{\alpha})}\boldsymbol{\phi}_N(\cdot)^\top \widehat{\boldsymbol{\theta}}_n\|_{L^\infty} \lesssim \left(\frac{n}{\log(n/\delta)}\right)^{-\frac{\nu-|\boldsymbol{\alpha}|}{2\nu+d}} \|f\|_{\mathcal{C}^\nu}^{\frac{2|\boldsymbol{\alpha}|+d}{2\nu+d}} \gamma \tag{11}$$

$$+ \left(\frac{n}{\log(n/\delta)}\right)^{-\frac{2\nu-2|\boldsymbol{\alpha}|}{2\nu+d}} \|f\|_{\mathcal{C}^\nu}^{\frac{4|\boldsymbol{\alpha}|+4d}{2\nu+d}} B, \tag{12}$$

*for $\|f\|_{\mathcal{C}^\nu}, \gamma, B \geq 1$.*

The preceding theorem establishes a connection with classical results in nonparametric statistics, which assert that, under the assumption of unit variance noise and $\|f\|_{\mathcal{C}^\nu} = 1$,

$$\|D^{(\boldsymbol{\alpha})}f(\cdot) - \widehat{f}^{\boldsymbol{\alpha}}_{\text{LPE},n}\|_{L^\infty} \asymp \left(\frac{n}{\log n}\right)^{-\frac{\nu - |\boldsymbol{\alpha}|}{2\nu + d}},$$

with probability one, where $f_{\text{LPE},n}$ denotes the Local Polynomial Estimator. Our result, for $\gamma = 1$, demonstrates that the uniform error of our estimator is asymptotic to $(n/\log n)^{-\frac{\nu - |\boldsymbol{\alpha}|}{2\nu + d}}$, independently on $B$, which only appears in the lower-order term.

## 5 Lower bound

As previously noted, the rate in $n$ obtained in the preceding theorem is asymptotically optimal, a result extensively documented in the non-parametric statistics literature. Nonetheless, to ensure that the aforementioned result is truly sharp, it is necessary to establish its optimality with respect to the constants that characterize the problem. Furthermore, given the framework of this work, we are interested in a high-probability finite-sample bound rather than an asymptotic one. In this section, we prove a lower bound, showing that our last result, Corollary 6, provides the best possible dependence in all variables.

**Theorem 10.** *Any algorithm outputting an estimation $D^{(\boldsymbol{\alpha})}\widehat{f}_n(\cdot)$ for $D^{(\boldsymbol{\alpha})}f(\cdot)$ suffers an error lower-bounded by*

$$\|D^{(\boldsymbol{\alpha})}f(\cdot) - D^{(\boldsymbol{\alpha})}\widehat{f}_n(\cdot)\| \geq \Omega\left(n^{-\frac{\nu - |\boldsymbol{\alpha}|}{2\nu + d}} \psi_0^{\frac{2|\boldsymbol{\alpha}| + d}{2\nu + d}} \sigma^{\frac{2\nu - 2|\boldsymbol{\alpha}|}{2\nu + d}}\right),$$

*w.p. at least $1/4$, on a problem instance satisfying assumptions 1 and 2 with $\|f(\cdot)\|_{\mathcal{C}^\nu} \leq \psi_0$.*

The idea of the theorem is to divide the set $[-1, 1]^d$ into disjoint regions and work on each of them separately. The key to doing this is finding functions that are both smooth and zero at the boundaries of the regions, with all their derivatives. In this way, we can construct the hard instance by arbitrarily combining these functions, one for each region, while still preserving smoothness and periodicity. Once this is done, the proof proceeds via a standard KL divergence argument.

## 6 Related works

The problem of nonparametric regression for a smooth function is one of the most classical problems in statistical learning. Among the numerous approaches introduced over the years, we summarize the one that best fits our problem.

**Parametric approaches**    Because smoothness is intrinsically local, establishing relations between nearby points is the most intuitive approach to our problem; thus, we apply local techniques. In the literature, several approaches based on piecewise polynomial regression (Sauve, 2009) have been studied, with different estimation schemes (Chaudhuri et al., 1994) and computational complexities (Lokshtanov et al., 2021). This family of methods would indeed work for estimating $f(\cdot)$, as well-known results Chaudhuri et al. (1994) show that smooth functions can be approximated locally by polynomials (think of Taylor series). Unfortunately, these approaches fail to provide an estimated function $\widehat{f}_n$ whose derivatives approximate those of $f$; in fact, restricting regression to multiple intervals yields discontinuous regression functions at the boundaries of these regions.

Perhaps the approach closest to our work is that of Maran & Szepesvári (2025). In their study, which addresses uniform bounds for linear regression with general feature maps, they employ the same sub-Gaussianity assumptions as ours to obtain sample-complexity results that hold across various scenarios. Their method (Theorem 11) can compete with the best regularizer for a given feature map, including the De la Vallée Poussin kernel used in this work. Nonetheless, due to lower-order terms, even in our scenario, their estimator does not achieve the optimal sample complexity for every triplet $d, \nu, \boldsymbol{\alpha}$, unlike ours. Furthermore, that paper focuses on function estimation rather than on its derivatives.

### 6.1 Non-Parametric approaches

As anticipated, the vast majority of the literature on non-parametric regression uses non-parametric estimators. Among the proposed methods, we identify four clusters that, to the best of our knowledge, encompass all the literature.

**Local Polynomial estimators** Historically, the first approaches that are able to get optimal approximation properties for a function and its derivatives in norm $L^\infty$ are Local Polynomial Estimators (LPE) (see Stone (1982; 1983); Delecroix & Rosa (1996); De Brabanter et al. (2013) and pages 34-42 from Tsybakov (2009) for a survey). This kind of estimator extends the Nadaraya–Watson (NW) family (see page 31 Tsybakov (2009)), achieving, in the asymptotic case, the same guarantee of our algorithm 7. Like our DUPA, LPE is able to estimate the derivatives of the target function with optimal order, as proved in the seminal papers Stone (1982; 1983). However, the estimator does *not* enjoy the plug-in property: the bandwidth, a crucial hyperparameter that characterizes the learning procedure, must be adjusted differently to estimate different derivatives.

**Splines** As we have seen, a smooth function can be well approximated by its Taylor polynomials built on several regions that cover the domain, but unfortunately, this type of estimator function is not even continuous. A solution for this issue can be found in Splines (Wahba, 1990; Quarteroni et al., 2010). Splines are, in brief, locally polynomial functions with a fixed degree of smoothness at the boundaries of the intervals. Several types of splines were introduced (Wahba, 1990; Marsh & Cormier, 2001; Acharjee & Das, 2022; Wang, 2011), always with the aim of obtaining an estimated function $\widehat{f}_n$ that is smooth on the whole domain. Smoothing Splines are the most studied in the theory of nonparametric regression. Still, to the best of our knowledge, no sample complexity results for $L^\infty$ error over a class of smooth functions have been shown. Current results (Stone, 1985; Zhou & Wolfe, 2000; Li & Ruppert, 2008; Claeskens et al., 2009; Charnigo et al., 2011) deal only with the $L^2$ error. These kinds of estimators are, in general, not plug-in and require the target function to be $(m+1)-$continuously differentiable in order to estimate the $m-$th derivative.

**Finite difference methods** Finite Difference (FD Dai et al. (2018); Liu & De Brabanter (2018)) methods offer a more "algebraic" and computationally direct alternative to Local Polynomial Estimation (LPE) for derivative estimation. While LPE fits a surface to data, FD approximates derivatives by taking weighted differences of function values at discrete points. Like Splines, these methods achieve order-optimal sample complexity, but only for the $L^2$ error. In general, these estimators do not enjoy the plug-in property and can require up to smoothness of order five to estimate the first derivative Wang & Lin (2015).

**RKHS** Recently, Liu & Li (2023) was able to prove that a modified version of Kernel ridge regression achieves quasi-optimal approximation $L^2$ of the derivatives under a finite-time analysis. Their algorithm enjoys the plug-in property, which, in the context of Kernelized Ridge Regression, means that the $\lambda$ parameter can be chosen once for all the derivatives. Precisely, to estimate a derivative of order $\alpha$ their estimator requires the function to be differentiable up to order $\alpha + 1/2$, and their sample complexity scales as $(n/\log n)^{-\frac{\nu'-|\alpha|}{2\nu'+d}}$, for every $\nu' < \nu$. Technically, this rate is suboptimal, even if it can be made arbitrarily close to the optimal one.

## 6.2 Comparison with LPE

As we have seen, the only algorithms that match our theoretical guarantees in $L^\infty$ are the LPE from the nonparametric statistics literature (Tsybakov, 2009). While our bound in Theorem 7 is valid for every $n$ and for constants that can be exactly computed, the performance of LPE depends on a constant $\lambda_0^{-1}$, where $\lambda_0$ is only bounded away from zero asymptotically (Tsybakov (2009) Lemma 1.4). On the other hand, an advantage of LPE is that their guarantee holds for $\{x_i\}_{i=1}^n$ that are uniformly distributed, while our result requires a more peculiar distribution. Arguably, the difference between the two approaches mostly concerns computational complexity.

**Different notions of computational complexity.** The efficiency of an algorithm must be measured in terms of time complexity (the number of operations required) and space complexity (the memory footprint incurred), both of which manifest differently during the training and prediction (inference) phases. Training complexity measures the resources required to optimize model parameters on a given dataset; this phase is typically computationally intensive for parametric methods and less intensive for nonparametric methods. In contrast, prediction complexity determines the resources required to process a new query after the model is deployed, and it is typically higher for non-parametric methods. While certain applications can tolerate computationally demanding training cycles, the constraints on prediction are often significantly more stringent. This asymmetric demand is particularly evident in real-time streaming applications, edge computing, and high-frequency trading, where algorithms must be both exceptionally fast and light during the prediction phase. In high-frequency trading, for instance, financial decisions must be executed within microseconds,

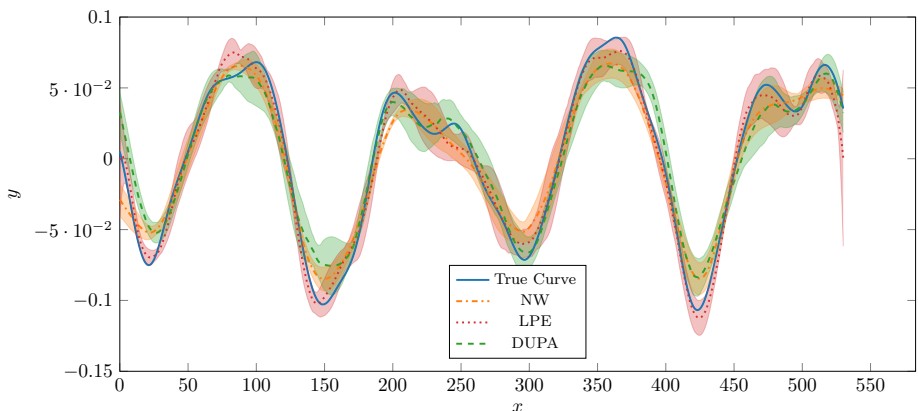

Figure 2: True unknown function $f(\cdot)$ used in the experiments and $95\%$ confidence regions for the predictions $f_n(\cdot)$ generated by three algorithms.

making sub-millisecond prediction latency and a minimal memory footprint strict prerequisites for maintaining a competitive edge.

**Comparison on computational complexity.** To compare our algorithm with LPE, we fix a number $n$ of training and $m$ of prediction samples. Results are summarized in table 1. (DUPA) There are only two parts of algorithm 1 that are computationally heavy: finding the optimal design at line 4 and solving the linear regression problem at line 19. The former step can be performed in $kN_d^2$ steps (Lattimore et al. (2020)), whereas the latter is well-known to require $nN_d^2 + N_d^3$ (the computational complexity of linear regression). Replacing $k = 1/\varepsilon = \mathcal{O}(N)$, we get a time complexity of $\mathcal{O}(nN_d^2 + N_d^3 + mN_d)$. For the optimal choice $N_d = \mathcal{O}(n^{\frac{d}{2\nu+d}})$. This number leads to $\mathcal{O}(n^{\frac{2\nu+3d}{2\nu+d}} + mn^{\frac{d}{2\nu+d}})$. The space complexity in training corresponds to storing the design matrix, which means $\mathcal{O}(N_d) = \mathcal{O}(n^{\frac{2d}{2\nu+d}})$, while the one in prediction to $\mathcal{O}(n^{\frac{d}{2\nu+d}})$, as the vector of estimated $\widehat{\theta}_n$ is sufficient. (NW/LPE) Both algorithms, like many non-parametric methods, are learned through lazy learning. That is, nothing is done in the training phase, and all samples are cycled through for every prediction. This leads to a time complexity of $\mathcal{O}(mn)$ and a space complexity of $\mathcal{O}(n)$, both in training and prediction. The algorithm of Liu & Li (2023) enjoys the usual computational/space complexity of kernel-ridge regression.

Our algorithm is always faster in the prediction case, as $\frac{d}{2\nu+d} < 1$. Moreover, the time complexity is less to that of LPE provided that $n^{\frac{2d}{2\nu+d}} < m$, which holds if either we have many more predictions than training samples or if $n \approx m$ and $\nu > d/2$. This occurs in several realistic situations in which the function $f(\cdot)$ arises from a physical process. Thermal and electromagnetic phenomena are governed by the heat equation and the Laplace-Poisson equation, respectively (Sobolev, 1964; Tikhonov & Samarskii, 2013; Salsa & Verzini, 2022). Each of these is characterized by inherent smoothness properties, making the solution infinitely many times differentiable in the interior of the domain so that we can take $\nu = \infty$. For this favorable case, our computational complexity approaches the dream-like $\widetilde{\mathcal{O}}(n + m)$, while the space is polylogarithmic, as it is possible to choose $N_d = \mathcal{O}(\log(n)^d)$. To close this section, we prove a novel result: no algorithm can achieve a space complexity lower than that of DUPA in the prediction phase.

**Theorem 11.** *Any algorithm with optimal statistical complexity for a regression problem satisfying assumptions 1 and 2 must have a space complexity in the prediction of at least* $\Omega\left(n^{\frac{d}{2\nu+d}}\right)$.

## 7 Experiments

To empirically validate the results of our DUPA algorithm, we test it on a regression task of a smooth function. As this paper is concerned about the case of $f(\cdot)$ being periodic, we focus on a common real case in which functions appear that are endowed with some kind of periodicity, that is, the one of audio signals (Purwins et al., 2019). In particular, our target function $f(\cdot)$ has been extracted from the signal of the song *Houdini* ©(Lipa et al., 2023) in the following way.

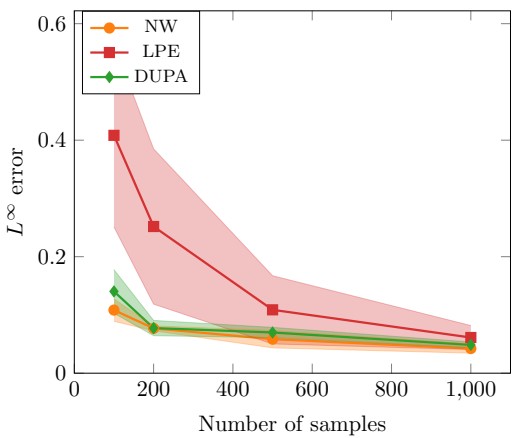
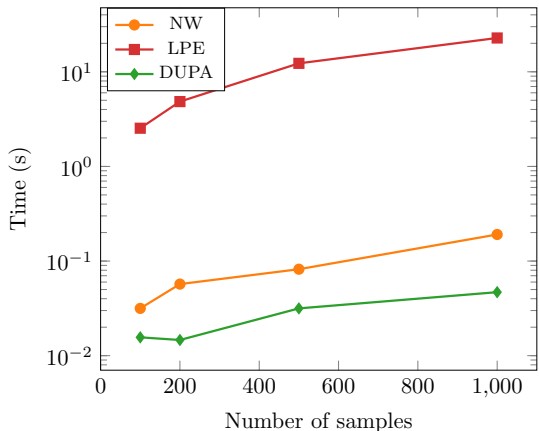

Figure 3: Left: $L^\infty$ error of each of the algorithm, each averaged over 5 random seeds, and with shaded regions representing 95% coverage confidence intervals for the estimation. Right: log-scale plot of the running time of each algorithm, compared to the number $n$ of samples.

The audio signal, originally comprising 8.2 million time steps, was too complex for direct use in our regression task. To prepare the signal for our experiment, we divided the waveform into intervals of length between 500 and 1000 samples. These intervals were carefully selected to ensure periodicity at their boundaries by taking only those that start and end with values close to zero (which is possible for audio signals, where the waveform naturally oscillates around zero). Then, one interval was used for hyperparameter tuning of the algorithms, and another for testing their performance. The plot of the test function is shown in Figure 2 as the blue solid line. As far as the noise of the observation is concerned, we have always used a zero-mean Gaussian of standard deviation $0.1$.

**Algorithms and results** In this numerical experiment, we have compared our algorithm DUPA with the previously introduced Nadaraya–Watson (NW) and the Local Polynomial Estimators (LPE). Kernel Ridge Regression is not present since, even if Liu & Li (2023) prove that it can estimate the derivatives uniformly at an optimal order, its computational complexity is so bad that it is not feasible to perform the experiment. We evaluated the performance of our DUPA algorithm and the baseline methods across four different sample sizes: $n = 100, 200, 500, 1000$. Figure 3 (left) shows that while for $n = 1000$ all algorithms are able to estimate $f(\cdot)$ almost perfectly, the error of our algorithm DUPA decreases much faster than the one of LPE, and similarly to the one of NW. As the other panel 3 (right) shows, DUPA obtains the best running time across the algorithms, outperforming LPE by orders of magnitude. In this experiment, we have taken the same values for $n$, while $m \approx 550$, as the length of the axis of figure 2, which has been rescaled to $[-1, 1]$ in the simulation. Although both LPE and NW share the same theoretical computational complexity of $\mathcal{O}(nm)$, in practice, LPE is significantly slower because it requires solving a small linear regression for each prediction sample. Although the size of these regressions is negligible relative to $n$, the additional computations incur significant overhead, leading to a noticeable increase in running time.

# 8 Conclusions

In this work, we revisited the classical problem of nonparametric regression through the lens of modern machine learning requirements. While uniform approximation guarantees for smooth functions are a cornerstone of nonparametric statistics, existing methods achieving optimal rates typically rely on nonparametric estimators whose computational and memory costs scale unfavorably with the sample size. This mismatch limits their applicability in contemporary learning scenarios where fast prediction, finite memory, and non-asymptotic guarantees are essential.

We showed that these limitations are not inherent to the problem itself. By combining tools from harmonic analysis, optimal experimental design, and finite-sample concentration, we introduced a parametric estimator that achieves minimax-optimal uniform convergence rates for both the target function and all its derivatives. Our guarantees hold in finite samples, are sharp with respect to all problem-dependent constants, and extend naturally to second-order

(Bernstein-type) bounds that adapt to the variance of the noise. Beyond statistical optimality, our approach leads to substantial gains in computational and memory efficiency. Unlike classical nonparametric estimators, the resulting predictor is lightweight at inference time, requiring storage only proportional to the number of parameters. We complemented our upper bounds with a matching lower bound, showing that this prediction-time memory complexity is information-theoretically optimal among all statistically optimal estimators.

Taken together, our results demonstrate that parametric methods, when carefully designed to respect the structure of smooth function classes, can match the statistical performance of nonparametric approaches while being fundamentally more compatible with the constraints of modern machine learning systems. We believe this perspective opens new avenues for bridging nonparametric statistics with areas such as bandit optimization, reinforcement learning, and continuous control, where uniform guarantees and computational efficiency are simultaneously required.

### 8.1 Future works

While this paper has focused on the classical notion of smoothness used in most works on Sobolev spaces, it is of paramount importance to generalize these results to broader classes of smooth function spaces. In particular, recent analyses have begun to investigate spaces of functions with *dominating mixed smoothness* Triebel (2019) to mitigate the curse of dimensionality. Our sample complexity guarantees show that the error scales as $n^{-\frac{\nu - |\boldsymbol{\alpha}|}{2\nu + d}}$. While optimal, this decay becomes very slow in high dimension $d$. Under the dominating mixed smoothness assumption, however, it may be possible to obtain bounds that are less affected by the dimensionality of the domain.

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

# A   Table of Notation

| | |
|---|---|
| $d$ | Dimension of the space, e.g. $[-1,1]^d$ |
| $n$ | number of samples available for the algorithm |
| $\sigma$ | sub-Gaussianity constant |
| $\mathbb{R}$ | Set of real numbers |
| $\nu$ | Index of space of differentiable functions $\mathcal{C}^\nu(\mathcal{X})$ |
| $\mathcal{C}^\nu(\mathcal{X})$ | Space of differentiable functions $\mathcal{C}^\nu(\mathcal{X})$ for some $\mathcal{X} \subset \mathbb{R}^d$ |
| $\nu_*$ | $\lceil \nu - 1 \rceil$ |
| $L_\nu(f)$ | Lipschitz constant of $f$ w.r.t. the index $\nu$ |
| $\alpha/\boldsymbol{\alpha}$ | index/multi-index of the derivative |
| $f^{(\alpha)}$ | derivative of a univariate function |
| $D^{(\boldsymbol{\alpha})}f$ | multi-index derivative for a multivariate function |
| $|\boldsymbol{\alpha}|$ | norm of the multiindex, corresponding to $\|\boldsymbol{\alpha}\|_1 = |\boldsymbol{\alpha}_1| + \cdots + |\boldsymbol{\alpha}_d|$ |
| $\| \cdot \|_{L^\infty}$ | supremum norm of a function, $\|f\|_{L^\infty} = \sup |f|$ |
| $\| \cdot \|_{L^2}$ | $= \sqrt{\int_\Omega f(x)^2 \, dx}$ for a function $f$ |
| $\| \cdot \|_{\mathcal{C}^\nu}$ | norm over $\mathcal{C}^\nu$ |
| $\mathbb{T}_N$ | Space of trigonometric polynomial of degree not exceeding $N$ |
| $T_N(\cdot)$ | Element of $\mathbb{T}_N$ |
| $\phi_N(\cdot)$ | Fourier (sin-cos) feature map |
| $\mathcal{F}_N[f](\cdot)$ | Fourier series of order $N$ associated to $f$ |
| $D_N(\cdot)$ | Dirichlet Kernel |
| $V_N(\cdot)$ | De la Valèe Poussin Kernel |
| $\beta_+ V_N^+(\cdot)$ $\beta_- V_N^-(\cdot)$ | $-$ Decomposition of the kernel $V_N(\cdot)$ (see (6)) |
| $N_d$ | $\binom{N+d}{N}$ |
| $\Sigma$ | Design matrix |

**Notation (Part 2)**

| | |
|---|---|
| $\mathcal{O}(\cdot)$ | Order of something, ignoring constants |
| $\widetilde{\mathcal{O}}(\cdot)$ | Order of something, ignoring constants and logarithms |
| $k$ | See algorithm 1 |
| $\varepsilon$ | See algorithm 1 |
| $n_{\text{tot}}$ | See algorithm 1 |
| $\text{supp}(\rho)$ | Support of a probability distribution $\rho(\cdot)$ (intersection of all closed sets of probability one) |
| $\mathcal{N}(x; \mu, \sigma)$ | Gaussian density function of parameters $\mu, \sigma$ evaluated in $x$ |
| $\Psi(\cdot)$ | Standard mollifier (see equation 52) |

## B    Proofs of section 3

**Theorem 4.** *Let $f \in \mathcal{C}_p^\nu([-1,1])$ and $\alpha \in \{0, 1, \dots \nu_*\}$. Then, $V_N * f(\cdot) \in \mathbb{T}_N$, and we have for a constant $K > 0$,*

$$\|f^{(\alpha)}(\cdot) - (V_N * f)^{(\alpha)}(\cdot)\|_{L^\infty} \le K(2/N)^{\nu - \alpha}\|f\|_{\mathcal{C}^\nu}.$$

*Proof.* The proof largely builds on Theorem 12. Indeed, for $\alpha = 0$ we have exactly the former result, while for $\alpha > 0$ we note that, by the properties of convolution, which commutes with differentiation, we have, $(V_N * f)^{(\alpha)} = V_N * (f^{(\alpha)})$; therefore,

$$\|f^{(\alpha)} - (V_N * f)^{(\alpha)}\|_{L^\infty} = \|f^{(\alpha)} - V_N * (f^{(\alpha)})\|_{L^\infty}.$$

At this point, we note that the function $f^{(\alpha)} \in \mathcal{C}^{\nu - \alpha}([-1, 1])$, as $f \in \mathcal{C}_p^\nu([-1, 1])$. Therefore, we can apply theorem 12 for $\nu' = \nu - \alpha$ to ensure

$$\|f^{(\alpha)} - (V_N * f)^{(\alpha)}\|_{L^\infty} \le K(2/N)^{\nu + \alpha}\|f^{(\alpha)}\|_{\mathcal{C}^{\nu - \alpha}}.$$

Bounding $\|f^{(\alpha)}\|_{\mathcal{C}^{\nu - \alpha}}$ by its definition, we can complete the proof:

$$\|f^{(\alpha)}\|_{\mathcal{C}^{\nu - \alpha}} = \max \left\{ \max_{\beta \le \nu_* - \alpha} \|D^\beta f^{(\alpha)}\|_{L^\infty}, L_{\nu - \alpha}(f^{(\alpha)}) \right\}$$

$$= \max \left\{ \max_{\beta \le \nu_* - \alpha} \|D^{\beta + \alpha} f\|_{L^\infty}, L_\nu(f) \right\}$$

$$= \max \left\{ \max_{\alpha \le \beta \le \nu_*} \|D^\beta f\|_{L^\infty}, L_\nu(f) \right\}$$

$$\le \|f\|_{\mathcal{C}^\nu}.$$

$\square$

**Theorem 12.** *(Theorem 2 in Maran et al. (2024b)) Let $f \in \mathcal{C}_p^\nu([-1, 1])$. Then, $V_N * f \in \mathbb{T}_N$, and we have*

$$\|f(\cdot) - V_N * f(\cdot)\|_{L^\infty} \le K_1 \inf_{T_{N/2} \in \mathbb{T}_{N/2}} \|T_{N/2}(\cdot) - f(\cdot)\|_{L^\infty} \le K_2(2/N)^\nu\|f\|_{\mathcal{C}^\nu},$$

*where $\mathbb{T}_N$ denotes the space of trigonometric polynomials of degree not higher than $N$ and $K_1, K_2$ are universal constants.*

## C    (Quasi-)Optimal Design

Algorithm 1 requires a quasi-optimal design (Kiefer & Wolfowitz, 1960) to choose which points to sample. To this aim, we recall a result by (Lattimore et al., 2020, Theorem 4.3).

**Theorem 13.** *(Lattimore et al. (2020), Theorem 4.4) Suppose $\mathcal{X} \subset \mathbb{R}^N$ is a compact set spanning $\mathbb{R}^N$. We can find a probability distribution $\rho$ on $\mathcal{X}$ such that $|supp(\rho)| \leq 4N \log\log(N)$ and, once defined*

$$\Sigma = \mathop{\mathbb{E}}_{\boldsymbol{x} \sim \rho} [\boldsymbol{x}\boldsymbol{x}^\top],$$

*we have, for all $\boldsymbol{x} \in \Omega$, $\|\boldsymbol{x}\|_{\Sigma^{-1}}^2 \leq 2N$ (here the notation $\|\boldsymbol{x}\|_{\Sigma^{-1}}$ stands for $\sqrt{\boldsymbol{x}^\top\Sigma^{-1}\boldsymbol{x}}$).*

The distribution $\rho$ defined in Theorem 13 is called *quasi-optimal design* for the least square problem. "Quasi" in the previous definition is because the actual optimal design satisfies $\boldsymbol{x} \in \Omega$, $\|\boldsymbol{x}\|_{\Sigma^{-1}}^2 \leq N$; unfortunately, the latter result would require a larger support for $\rho$, and it is not suitable for our problem henceforth.

## D  Proofs of section 4

**Theorem 5.** *Run* DUPA *Algorithm 1 for a choice $N$ such that $16N \log\log(N) < n$. Under Assumptions 1,2 for $d = 1$, with probability at least $1 - \delta$, the output satisfies, for all $\alpha = \{0, \dots \nu_*\}$*

$$\|f^{(\alpha)}(\cdot) - \phi_N^{(\alpha)}(\cdot)^\top \widehat{\theta}_n\|_{L^\infty} \lesssim N^\alpha \left( \frac{\|f\|_{\mathcal{C}^\nu}}{N^\nu} + \sqrt{\frac{N \log(n/\delta)}{n}}\sigma \right), \tag{8}$$

*for $\|f\|_{\mathcal{C}^\nu}, \sigma \geq 1$.*

*Proof.* **(Part 1: Expected target and sub-Gaussianity)** We start showing that the average target of the regression problem is exactly linear in the feature map. Indeed, we have, for every $i = 1, \dots n$,

$$\mathbb{E}[y_i] = \mathbb{E}[\beta_+ y_i^+] - \mathbb{E}[\beta_- y_i^-] \tag{13}$$
$$= \beta_+ \mathbb{E}[y_i^+] - \beta_- \mathbb{E}[y_i^-] \tag{14}$$
$$= \beta_+ \mathbb{E}[f(x_i + \eta_+)] - \beta_- \mathbb{E}[f(x_i + \eta_-)] \tag{15}$$
$$= \beta_+ V_N^+ * f(x_i) - \beta_- V_N^+ * f(x_i) \tag{16}$$
$$= V_N * f(x_i). \tag{17}$$

Here, step 16 follows from 7. Now, let us focus on the sub-Gaussianity. We have

$$y_i - V_N * f(x_i) = \beta_+ y_i^+ - \beta_- y_i^- - V_N * f(x_i) \tag{18}$$
$$= (\beta_+ y_i^+ - \beta_+ \mathbb{E}[y_i^+]) - (\beta_- y_i^- - \beta_- \mathbb{E}[y_i^-]) \tag{19}$$
$$= \beta_+ \underbrace{(y_i^+ - \mathbb{E}[y_i^+])}_{T1} - \underbrace{\beta_-(y_i^- - \mathbb{E}[y_i^-])}_{T2}. \tag{20}$$

Consider $(T1)$. This term writes as

$$y_i^+ - \mathbb{E}[y_i^+] = y_i^+ - \mathbb{E}[y_i^+|\eta_+] + \mathbb{E}[y_i^+|\eta_+] - \mathbb{E}[y_i^+]$$
$$= y_i^+ - \mathbb{E}[f(x_i + \eta_+)|\eta_+] + \mathbb{E}[f(x_i + \eta_+)|\eta_+] - V_N^+ * f(x_i)$$

where the both terms have zero mean. By assumption 1 the first term is $\sigma-$subgaussian, and by the fact that $f$ is bounded in $[0, 1]$ the second term is also $1-$sugbaussian. The same reasoning trivially applies to $(T2)$, which is also $(\sigma + 1)-$sub-Gaussian. Therefore, looking at the sum of two independent sub-Gaussians, we have that $y_i - V_N * f(x_i)$ is $\sigma'-$sub-Gaussian with

$$\sigma' = (\beta_+^2 + \beta_-^2)^{1/2}(\sigma + 1) \leq (|\beta_+| + |\beta_-|)(\sigma + 1) = \Lambda_1(\sigma + 1).$$

Where $\Lambda_1$ is the constant of equation 5. Having proved that $y_i$ is unbiased w.r.t. $V_N * f(x_i)$ allows us to ensure the existence of some $\theta_\star \in \mathbb{R}^{2N+1}$ (as $\mathbb{T}_N$ is a vector space of dimension $2N + 1$, as clear from equation (2)) such that

$$\mathbb{E}[y_i] = V_N * f(x_i) = \phi_N(x_i)^\top \theta_\star,$$

as it follows from the first part of theorem 4 (the writing $\phi_N(x_i)^\top \theta_\star$ corresponds to saying that the function belongs to $\mathbb{T}_N$).

**(Part 2: The discretization makes no harm)** At this point, we can apply proposition 2 from Maran et al. (2024b), with $\boldsymbol{x}_t = \phi_N(x_t)$ and $\mathcal{X}' = \mathcal{C}_\varepsilon$, defined on line 3. In this way, $k = 1/\varepsilon$. The latter result proves, with probability $1 - \delta$,

$$\forall x \in \mathcal{C}_\varepsilon, \qquad |\phi_N(x)^\top \widehat{\theta}_n - \phi_N(x)^\top \theta_\star| \leq \sqrt{\log(k/\delta)} \sup_{x \in \mathcal{C}_\varepsilon} \|x\|_{\Sigma_n^{-1}} \sigma', \tag{21}$$

where

$$\Sigma_n := \sum_{i=1}^n \phi(x_i)\phi(x_i)^\top.$$

At this point, we have to generalize the previous bound to all points of $[-1, 1]$, even not belonging to $\mathcal{C}_\varepsilon$. To do this, we call $\Delta f_n := \phi_N(\cdot)^\top(\widehat{\theta}_n - \theta_\star) \in \mathbb{T}_N$. The fact that the function is a trigonometric polynomial allows us to apply Lagrange's theorem, which ensures that, for any couple of points $x_1, x_2$, there is $x' \in [x_1, x_2]$ such that

$$\Delta f_n(x_1) - \Delta f_n(x_2) = \Delta f_n^{(1)}(x')(x_1 - x_2). \tag{22}$$

Therefore, we have

$$\|\Delta f_n(\cdot)\|_{L^\infty} = \sup_{x_1 \in [-1,1]} |\Delta f_n(x_1)| \tag{23}$$

$$\overset{(22)}{=} \sup_{x_1 \in [-1,1]} \inf_{x_2 \in \mathcal{C}_\varepsilon} |\Delta f_n(x_2) - \Delta f_n^{(1)}(x')(x_1 - x_2)| \tag{24}$$

$$\leq \sup_{x_1 \in [-1,1]} \inf_{x_2 \in \mathcal{C}_\varepsilon} |\Delta f_n(x_2)| + |\|\Delta f_n^{(1)}\|_{L^\infty}(x_1 - x_2)| \tag{25}$$

$$\leq \sup_{x \in \mathcal{C}_\varepsilon} |\Delta f_n(x)| + \varepsilon \|\Delta f_n^{(1)}\|_{L^\infty}. \tag{26}$$

Here, (25) follows from bounding the derivative with its infinity norm, and the one (26) from choosing $x_2$ to be the nearest element of $x_1$ on $\mathcal{C}_\varepsilon$. Finally, we can bound $\|\Delta f_n^{(1)}\|_{L^\infty}$. Indeed, due to the fact that the function is a degree-$N$ trigonometric polynomial, by Theorem 17 we have

$$\|\Delta f_n^{(1)}\|_{L^\infty} \leq 4\pi N \|\Delta f_n(\cdot)\|_{L^\infty}., \tag{27}$$

which, once merged with the previous result, provides

$$\|\Delta f_n(\cdot)\|_{L^\infty} \leq \frac{1}{1 - 4\pi\varepsilon N} \sup_{x \in \mathcal{C}_\varepsilon} |\Delta f_n(x)|. \tag{28}$$

Merging equations (21) and (28), we get

$$\|\Delta f_n(\cdot)\|_{L^\infty} \leq \frac{1}{1 - 4\pi\varepsilon N} \sqrt{\log(k/\delta)} \sup_{x \in \mathcal{C}_\varepsilon} \|x\|_{\Sigma_n^{-1}} \sigma'.$$

**(Part 3: Optimal design)** Here, what is missing is to estimate $\|x\|_{\Sigma_n^{-1}}$. This can be done by line 4: indeed,

- $\rho$ is a quasi-optimal design for $\mathcal{X}_\phi$.

- The samples are collected according to $\lceil n_{\text{tot}} \rho(x) \rceil$, which dominates $n_{\text{tot}} \rho(x)$.

Therefore, from theorem 13, $\|x\|_{\Sigma_n^{-1}} \le \sqrt{2(2N+1)/n_{tot}}$. This result proves that

$$\|\Delta f_n(\cdot)\|_{L^\infty} \le \frac{1}{1 - 4\pi\varepsilon N} \sqrt{5\log(k/\delta)N/n_{tot}} \sigma'. \tag{29}$$

Moreover, Theorem 17 ensures that an analogous result also holds for the derivatives:

$$\forall \alpha \in \mathbb{N} \qquad \|\Delta f_n^{(\alpha)}(\cdot)\|_{L^\infty} \le \frac{(4\pi)^\alpha N^{1/2+\alpha}}{1 - 4\pi\varepsilon N} \sqrt{5\log(k/\delta)/n_{tot}} \sigma'. \tag{30}$$

**(Part 4)** At this point, it is just a matter of substituting the known constants and applying theorem 4. For any $\alpha \in \{0, \dots \nu_*\}$ we have

$$\|f^{(\alpha)} - \phi_N^{(\alpha)}(\cdot)^\top \widehat{\theta}_n\|_{L^\infty} \le \|f^{(\alpha)} - (V_N * f)^{(\alpha)}\|_{L^\infty} + \|(V_N * f)^{(\alpha)} - \phi_N^{(\alpha)}(\cdot)^\top \widehat{\theta}_n\|_{L^\infty} \tag{31}$$

$$= \|f^{(\alpha)} - (V_N * f)^{(\alpha)}\|_{L^\infty} + \|\Delta f_n^{(\alpha)}\|_{L^\infty} \tag{32}$$

$$\overset{\text{thm. 4}}{\le} K(2/N)^{\nu-\alpha} \|f\|_{\mathcal{C}^\nu} + \|\Delta f_n^{(\alpha)}\|_{L^\infty} \tag{33}$$

$$\overset{(30)}{\le} K(2/N)^{\nu-\alpha} \|f\|_{\mathcal{C}^\nu} + \frac{(4\pi)^\alpha N^{1/2+\alpha}}{1 - 4\pi\varepsilon N} \sqrt{5\log(k/\delta)/n_{tot}} \sigma', \tag{34}$$

where the first inequality comes from the triangular inequality, the second inequality from theorem 4, and the third one from equation (30), which we have obtained in this proof. At this point, we can replace $n_{tot} = \lfloor n/4 \rfloor$, $k = \varepsilon^{-1}$, $\sigma' = \Lambda_1 \sigma$ and $4\pi\varepsilon N = 1/2$ to get

$$\|f^{(\alpha)} - \phi_N^{(\alpha)}(\cdot)^\top \widehat{\theta}_n\|_{L^\infty} \le K(2/N)^{\nu-\alpha} \|f\|_{\mathcal{C}^\nu} + 2(4\pi)^\alpha N^{1/2+\alpha} \sqrt{5\log(\varepsilon^{-1}/\delta)/n_{tot}} \Lambda_1 \sigma',$$

which means, up to constant factors, replacing $\sigma'$ with $\sigma$ (this is allowed for $\sigma \ge 1$ taking into account that $\Lambda_1$ is a constant),

$$\|f^{(\alpha)} - \phi_N^{(\alpha)}(\cdot)^\top \widehat{\theta}_n\|_{L^\infty} \lesssim N^\alpha \left( \frac{\|f\|_{\mathcal{C}^\nu}}{N^\nu} + \sqrt{\frac{N\log(n/\delta)}{n}} \sigma \right). \tag{35}$$

Let us now bound the number of samples collected by the algorithm. The queries for the model happen at lines 13 and 14, so that the total number of samples is

$$2 \sum_{x \in \text{supp}(\rho)} \lceil n_{\text{tot}} \rho(x) \rceil \le 2n_{\text{tot}} + 2|\text{supp}(\rho)| \le 2n_{\text{tot}} + 8N \log\log(N).$$

By assumption, both $2n_{\text{tot}}$ and $8N \log\log(N)$ are less than $n/2$, so that this number is itself bounded by $n$ as required by the algorithm.

$\square$

## E  Proofs from section 4.1

In this section, we are going to generalize the main result of this paper, Theorem 5, to the case of functions with domain $[-1,1]^d$. Fourier series in $[-1,1]^d$ can be built in a similar way to the univariate setting Katznelson (2004). Still, this time, the space $\mathbb{T}_{N,d}$ of trigonometric polynomials of degree $N$ no longer has dimension $2N+1$, as we also need

to take into account the mixed terms between the $d$ variables. In fact, it was proved (see section 3 from Maran et al. (2024b)) that these trigonometric polynomials take the form

$$T_N(\boldsymbol{x}) = \boldsymbol{\phi}_d(\boldsymbol{x})^\top \boldsymbol{\theta}, \qquad \boldsymbol{\theta} \in \mathbb{R}^{N_d}, \ N_d = \binom{2N+d}{d}.$$

In fact, the latter number corresponds to the number of integer-valued vectors $\boldsymbol{v}$ such that $\|\boldsymbol{v}\|_1 \leq N$, and can be bounded by $(2N+1)^d$.

About the approximation properties, we have that both Theorem 3, the result to approximate the smooth function with trigonometric polynomials, and Theorem 4, the projection with the la Valée Poussin Kernel, are still valid (the proof of Theorem 4 is the same in dimension $d > 1$). Also, the results about this latter function maintain their validity, just that this time:

$$\|V_{N,d}(\cdot)\|_{L^1} \leq \Lambda_d, \tag{36}$$

a dimension-dependent constant $\Lambda_d$ whose expression can be found in Németh (2016).

**Theorem 14.** *Assume to run algorithm 1 for a choice $N$ such that $16N_d \log\log(N_d) < n$. With probability at least $1 - \delta$, the output satisfies, for all $|\boldsymbol{\alpha}| \leq \nu_*$,*

$$\|D^{(\boldsymbol{\alpha})} f(\cdot) - D^{(\boldsymbol{\alpha})} \boldsymbol{\phi}_d(\cdot)^\top \widehat{\theta}_n\|_{L^\infty} \lesssim N^{|\boldsymbol{\alpha}|} \left( \frac{\|f\|_{\mathcal{C}^\nu}}{N^\nu} + \sqrt{\frac{N^d \log(n/\delta)}{n}} \sigma \right),$$

*for $\|f\|_{\mathcal{C}^\nu}, \sigma \geq 1$.*

*Proof.* **(Part 1)** This part follows exactly as made in the analog part of the proof of Theorem 5, proving that

$$\mathbb{E}[y_i] = V_{N,d} * f(x_i) \tag{37}$$

and that $y_i - \mathbb{E}[y_i]$ is sub-Gaussian for

$$\sigma' \leq (|\beta_+| + |\beta_-|)(\sigma + 1) = \Lambda_d(\sigma + 1). \tag{38}$$

**(Part 2)** differently from the proof of theorem 5, the cardinality of the $\varepsilon$−cover here is $k := |\mathcal{C}_\varepsilon| = (2/\varepsilon)^d$. Therefore, applying proposition 2 from Maran et al. (2024b) results in

$$\forall x \in \mathcal{C}_\varepsilon, \qquad |\boldsymbol{\phi}_N(x)^\top \widehat{\boldsymbol{\theta}}_n - \boldsymbol{\phi}_N(x)^\top \boldsymbol{\theta}_\star| \leq \sqrt{\log(k/\delta)} \sup_{x \in \mathcal{C}_\varepsilon} \|x\|_{\Sigma_n^{-1}} \sigma', \tag{39}$$

where $\boldsymbol{\theta}_\star$ is such that, $\boldsymbol{\phi}_N(\cdot)^\top \boldsymbol{\theta}_\star = V_{N,d} * f(\cdot)$ (this is possible since $V_{N,d} * f \in \mathbb{T}_{N,d}$), and

$$\Sigma_n := \sum_{i=1}^n \boldsymbol{\phi}_d(x_i) \boldsymbol{\phi}_d(x_i)^\top.$$

At this point, we have to generalize the previous bound to all points of $[-1,1]^d$, even if they do not belong to $\mathcal{C}_\varepsilon$. To do this, we call $\Delta f_n := \boldsymbol{\phi}_d(\cdot)^\top (\widehat{\boldsymbol{\theta}}_n - \boldsymbol{\theta}_\star) \in \mathbb{T}_{N,d}$. The fact that the function is a trigonometric polynomial (that is always differentiable) allows us to apply Lagrange's theorem, which ensures that, for any couple of points $x_1, x_2$, there is $x'$ in the segment between the two points such that

$$\Delta f_n(x_1) - \Delta f_n(x_2) = \nabla[\Delta f_n](x')(x_1 - x_2). \tag{40}$$

Here, $\nabla$ denotes the gradient operator. Using this result, we have

$$\|\Delta f_n(\cdot)\|_{L^\infty} = \sup_{x_1 \in [-1,1]^d} |\Delta f_n(x_1)| \tag{41}$$

$$\overset{(40)}{=} \sup_{x_1 \in [-1,1]^d} \inf_{x_2 \in \mathcal{C}_\varepsilon} |\Delta f_n(\boldsymbol{x}_2) - \nabla[\Delta f_n](\boldsymbol{x}')^\top (x_1 - x_2)| \tag{42}$$

$$= \sup_{x_1 \in [-1,1]^d} \inf_{x_2 \in \mathcal{C}_\varepsilon} |\Delta f_n(x_2)| + |\nabla[\Delta f_n](x')^\top (x_1 - x_2)| \tag{43}$$

$$\leq \sup_{x_1 \in [-1,1]^d} \inf_{x_2 \in \mathcal{C}_\varepsilon} |\Delta f_n(x_2)| + \|\nabla[\Delta f_n](x')^\top\|_\infty \|x_1 - x_2\|_1 \tag{44}$$

$$\leq \sup_{x_1 \in [-1,1]^d} \inf_{x_2 \in \mathcal{C}_\varepsilon} |\Delta f_n(x_2)| + \|\|\nabla[\Delta f_n](\cdot)^\top\|_\infty\|_{L^\infty} \|x_1 - x_2\|_1 \tag{45}$$

$$\leq \sup_{x \in \mathcal{C}_\varepsilon} |\Delta f_n(x)| + \varepsilon \|\|\nabla[\Delta f_n](\cdot)^\top\|_\infty\|_{L^\infty}. \tag{46}$$

Here, in step 44, we have used the Cauchy-Schwartz inequality between norm one and norm infinity, (45) follows from bounding the $\|\|\nabla[\Delta f_n](x')^\top\|_\infty$ with its infinity norm, and the one (46) from choosing $x_2$ to be the nearest element of $x_1$ on $\mathcal{C}_\varepsilon$ w.r.t. the $1-$norm. At this point, theorem 18 ensures $\|\|\nabla[\Delta f_n](\cdot)^\top\|_\infty\|_{L^\infty} \leq 4\pi N \|\Delta f_n(\cdot)^\top\|_{L^\infty}$, which gives

$$\|\Delta f_n(\cdot)\|_{L^\infty} \leq \sup_{x \in \mathcal{C}_\varepsilon} |\Delta f_n(x)| + \varepsilon 4\pi N \|\Delta f_n(\cdot)^\top\|_{L^\infty}.$$

We rewrite this result as

$$\|\Delta f_n(\cdot)\|_{L^\infty} \leq \frac{1}{1 - 4\pi\varepsilon N} \sup_{x \in \mathcal{C}_\varepsilon} |\Delta f_n(x)|., \tag{47}$$

and merge it with equation (39), getting

$$\|\Delta f_n(\cdot)\|_{L^\infty} \leq \frac{1}{1 - 4\pi\varepsilon N} \sqrt{\log(k/\delta)} \sup_{x \in \mathcal{C}_\varepsilon} \|x\|_{\Sigma_n^{-1}} \sigma'. \tag{48}$$

(Part 3) This point corresponds to its analog for the univariate case, except for the fact that, having a feature map $\phi_N(\cdot)$ of dimension $N_d$, the quasi-optimal design gives $\|\phi_d(x)\|_{\Sigma_n^{-1}} \leq \sqrt{2N_d/n_{tot}}$. Replacing this quantity, we get

$$\|\Delta f_n(\cdot)\|_{L^\infty} \leq \frac{1}{1 - 4\pi\varepsilon N} \sqrt{2\log(k/\delta)N_d/n_{tot}}\sigma', \tag{49}$$

and, applying theorem 17,

$$\forall \boldsymbol{\alpha} \qquad \|D^{(\alpha)}\Delta f_n(\cdot)\|_{L^\infty} \leq \frac{(4\pi)^{|\boldsymbol{\alpha}|} N_d^{1/2} N^{|\boldsymbol{\alpha}|}}{1 - 4\pi\varepsilon N} \sqrt{2\log(k/\delta)/n_{tot}}\sigma'. \tag{50}$$

(Part 4) Repeating the same procedure of the univariate case with $\lesssim$, we ignore constant terms. For any $\alpha \in \{0, \ldots \nu_*\}$, it holds

$$\|D^{(\boldsymbol{\alpha})} f - D^{(\boldsymbol{\alpha})} \boldsymbol{\phi}_N(\cdot)^\top \widehat{\theta}_n\|_{L^\infty} \leq \|D^{(\boldsymbol{\alpha})} f - D^{(\boldsymbol{\alpha})} (V_{d,N} * f)\|_{L^\infty}$$
$$+ \|D^{(\boldsymbol{\alpha})} (V_{d,N} * f) - D^{(\boldsymbol{\alpha})} \boldsymbol{\phi}_N(\cdot)^\top \widehat{\theta}_n\|_{L^\infty}$$
$$= \|D^{(\boldsymbol{\alpha})} f - D^{(\boldsymbol{\alpha})} (V_{d,N} * f)\|_{L^\infty} + \|D^{(\boldsymbol{\alpha})} \Delta f_n\|_{L^\infty}$$
$$\lesssim N^{|\boldsymbol{\alpha}| - \nu} \|f\|_{\mathcal{C}^\nu} + \|D^{(\boldsymbol{\alpha})} \Delta f_n\|_{L^\infty}$$
$$\overset{(50)}{\leq} N^{|\boldsymbol{\alpha}| - \nu} \|f\|_{\mathcal{C}^\nu} + \frac{(4\pi)^{|\boldsymbol{\alpha}|} N_d^{1/2} N^{|\boldsymbol{\alpha}|}}{1 - 4\pi\varepsilon N} \sqrt{2 \log(k/\delta)/n_{tot}} \sigma'$$
$$\lesssim N^{|\boldsymbol{\alpha}| - \nu} \|f\|_{\mathcal{C}^\nu} + N_d^{1/2} N^{|\boldsymbol{\alpha}|} \sqrt{\frac{\log(n/\delta)}{n}} \sigma.$$

Replacing $\sigma'$ with $\sigma$ is allowed for $\sigma \geq 1$ taking into account that $\Lambda_d$ is a constant.

Once recalled that $N_d = \mathcal{O}(N^d)$, this proves the desired bound:

$$\|D^{(\boldsymbol{\alpha})} f - D^{(\boldsymbol{\alpha})} \boldsymbol{\phi}_N(\cdot)^\top \widehat{\theta}_n\|_{L^\infty} \lesssim N^{|\boldsymbol{\alpha}|} \left( \frac{\|f\|_{\mathcal{C}^\nu}}{N^\nu} + \sqrt{\frac{N^d \log(n/\delta)}{n}} \sigma \right).$$

To conclude the proof, it is sufficient to prove that the number of samples required is at most $n$. The queries for the model happen at lines 13 and 14. Moreover, theorem 13 ensures that the optimal design satisfies $|\mathrm{supp}(\rho)| \leq 8 N_d \log\log(N_d)$, so that the total number of samples is

$$2 \sum_{x \in \mathrm{supp}(\rho)} \lceil n_{\mathrm{tot}} \rho(x) \rceil \leq 2 n_{\mathrm{tot}} + 2 |\mathrm{supp}(\rho)| \leq 2 n_{\mathrm{tot}} + 8 N_d \log\log(N_d).$$

By assumption, both $2 n_{\mathrm{tot}}$ and $8 N_d \log\log(N_d)$ are less than $n/2$, so that this number is itself bounded by $n$ as required by the algorithm. $\qquad\square$

**Theorem 7.** *Run* DUPA *algorithm 1 for*

$$N = \left( \frac{n}{\log(n/\delta)} \right)^{\frac{1}{2\nu + d}} \|f\|_{\mathcal{C}^\nu}^{\frac{2}{2\nu + d}} \sigma^{-\frac{2}{2\nu + d}}.$$

*Under assumptions 1,2 we have with probability at least $1 - \delta$, for all $|\boldsymbol{\alpha}| \leq \nu_*$,*

$$\|D^{(\boldsymbol{\alpha})} f(\cdot) - D^{(\boldsymbol{\alpha})} \boldsymbol{\phi}_N(\cdot)^\top \widehat{\boldsymbol{\theta}}_n\|_{L^\infty} \lesssim \left( \frac{n}{\log(n/\delta)} \right)^{-\frac{\nu - |\boldsymbol{\alpha}|}{2\nu + d}} \|f\|_{\mathcal{C}^\nu}^{\frac{2|\boldsymbol{\alpha}| + d}{2\nu + d}} \sigma^{\frac{2\nu - 2|\boldsymbol{\alpha}|}{2\nu + d}}, \tag{10}$$

*for $\|f\|_{\mathcal{C}^\nu}, \sigma \geq 1$.*

*Proof.* It is sufficient to replace

$$N = \left( \frac{n}{\log(n/\delta)} \right)^{\frac{1}{2\nu + d}} \|f\|_{\mathcal{C}^\nu}^{\frac{2}{2\nu + d}} \sigma^{-\frac{2}{2\nu + d}},$$

in the previous result. Note that the assumption is automatically satisfied as for this choice of $N$

$$N_d = \widetilde{\mathcal{O}}(n^{\frac{d}{2\nu + d}}) < \widetilde{\mathcal{O}}(n^1).$$

$\qquad\square$

### E.1 Bernstein bound

**Theorem 15.** *Under assumption 8. Assume to run algorithm 1 for a choice $N$ such that $16N_d \log\log(N_d) < n$. With probability at least $1 - \delta$, the output satisfies, for all $|\boldsymbol{\alpha}| \leq \nu_*$,*

$$\|D^{(\boldsymbol{\alpha})}f - D^{(\boldsymbol{\alpha})}\boldsymbol{\phi}_N(\cdot)^\top \widehat{\theta}_n\|_{L^\infty} \lesssim N^{|\boldsymbol{\alpha}|} \left( \frac{\|f\|_{\mathcal{C}^\nu}}{N^\nu} + \sqrt{\frac{N^d \log(N/\delta)}{n}}\gamma + \frac{N^d \log(N/\delta)}{n}B \right),$$

*for $\|f\|_{\mathcal{C}^\nu}, \gamma, B \geq 1$.*

*Proof.* Most of the proof goes exactly as in the proof of Theorem 14. We write here only the points that need to be changed applying theorem 16.

**(Part 1)** The same reasoning proves that $y_i - \mathbb{E}[y_i]$ is bounded by $\Lambda_d(B+1)$ and its variance does not exceed $\Lambda_d^2(\gamma+1)^2$. As the theorem assumes sufficiently large $\gamma$ and $B$, and $\Lambda_d$ is a constant, these terms are $\lesssim B, \gamma^2$ respectively.

**(Part 2)** as in the previous proof, we take a $\varepsilon-$cover $\mathcal{C}_\varepsilon$ of $[-1,1]^d$, whose cardinality is $k := |\mathcal{C}_\varepsilon| = (2/\varepsilon)^d$. We then apply theorem 16, with $\mathcal{V} = \{\boldsymbol{\phi}_N(x) : x \in C_\varepsilon\}$, which proves, with probability at least $1 - \delta$,

$$\forall x \in \mathcal{C}_\varepsilon, \qquad |\boldsymbol{\phi}_N(x)^\top \widehat{\boldsymbol{\theta}}_n - \boldsymbol{\phi}_N(x)^\top \boldsymbol{\theta}_\star| \leq \sqrt{\frac{4N_d\Lambda_d^2\gamma^2 \log(k/\delta)}{n_{\text{tot}}}} + \frac{4N_d\Lambda_d B \log(k/\delta)}{3n_{\text{tot}}}, \tag{51}$$

where $\boldsymbol{\theta}_\star$ is such that, $\boldsymbol{\phi}_N(\cdot)^\top\boldsymbol{\theta}_\star = V_{N,d} * f(\cdot)$ (this is possible since $V_{N,d} * f \in \mathbb{T}_{N,d}$).

The rest of part 2 and part 3 follow exactly as before.

**(Part 4)** For any $\alpha \in \{0, \ldots \nu_*\}$, the usual bias-variance decomposition gives

$$\|D^{(\boldsymbol{\alpha})}f - D^{(\boldsymbol{\alpha})}\boldsymbol{\phi}_N(\cdot)^\top\widehat{\theta}_n\|_{L^\infty} \lesssim N^{|\boldsymbol{\alpha}|-\nu}\|f\|_{\mathcal{C}^\nu} + N^{|\boldsymbol{\alpha}|}\sqrt{\frac{N_d \log(N/\delta)}{n}}\gamma + N^{|\boldsymbol{\alpha}|}\frac{N_d B \log(N/\delta)}{n}.$$

Once recalled that $N_d = \mathcal{O}(N^d)$, this proves the desired bound:

$$\|D^{(\boldsymbol{\alpha})}f - D^{(\boldsymbol{\alpha})}\boldsymbol{\phi}_N(\cdot)^\top\widehat{\theta}_n\|_{L^\infty} \lesssim N^{|\boldsymbol{\alpha}|} \left( \frac{\|f\|_{\mathcal{C}^\nu}}{N^\nu} + \sqrt{\frac{N^d \log N}{n}}\gamma + \frac{N^d \log(N/\delta)}{n}B \right).$$

The rest of the proof follows as before. $\square$

## F  Proofs from section 5

Before presenting the actual proof, we need to introduce some lemmas. We start with a general result about Gaussian distributions.

**Lemma 1.** *Let $\mathcal{N}(x; \nu, \sigma) := \frac{1}{\sqrt{2\pi}\sigma}e^{-(x-\mu)^2/(2\sigma^2)}$, the normal density function. We have, for all $v \in \mathbb{R}$,*

$$\int_{\mathbb{R}} \mathcal{N}(x; \mu, \sigma) \log \frac{\mathcal{N}(x; \mu, \sigma)}{\mathcal{N}(x + v; \mu, \sigma)} \, dx = \frac{v^2}{2\sigma^2}.$$

*Proof.*

$$\begin{aligned}
\int_{\mathbb{R}} \mathcal{N}(x; \mu, \sigma) \log \frac{\mathcal{N}(x; \mu, \sigma)}{\mathcal{N}(x + \mu; \mu, \sigma)} \, dx &= \int_{\mathbb{R}} \mathcal{N}(x; \mu, \sigma) \left( \frac{(x + v - \mu)^2}{2\sigma^2} - \frac{(x - \mu)^2}{2\sigma^2} \right) \, dx \\
&= \frac{1}{2\sigma^2} \int_{\mathbb{R}} \mathcal{N}(x; \mu, \sigma) \left( 2v(x - \mu) + v^2 \right) \, dx \\
&= \frac{v^2}{2\sigma^2} + \frac{2v}{2\sigma^2} \int_{\mathbb{R}} \mathcal{N}(x; \mu, \sigma)(x - \mu) \, dx.
\end{aligned}$$

The last integral gives exactly zero, as corresponds to the mean of $\mathcal{N}(\cdot; \mu, \sigma)$ minus $\mu$. This ends the proof. $\square$

The next lemmas will be based on the properties of the standard mollifier, a function with many applications in mathematical analysis.

## F.1 The standard mollifier

Let $d \in \mathbb{N}$. We call standard mollifier Evans (2022) the function $\mathbb{R}^d \to \mathbb{R}$

$$\Psi(x) := \begin{cases} e^{-\frac{1}{\|x\|_2^2 - 1}} & \|x\|_2^2 < 1 \\ 0 & \|x\|_2^2 \geq 1. \end{cases} \tag{52}$$

This function is well-known Evans (2022) for being infinitely times differentiable with compact support $B_1(0)$. Apart from this property, we need to prove this very simple lemma.

**Lemma 2.** *Let $\Psi(\cdot)$ be defined as in equation* (52). *Then, for all $\nu$, there is a constant $c_\nu > 0$ such that*

$$\forall |\boldsymbol{\alpha}| \leq \nu_* \qquad \|D^{(\boldsymbol{\alpha})}\Psi(\cdot)\|_{L^\infty} \geq c_\nu.$$

*Proof.* We take straightforwardly,

$$c_\nu = \min_{|\boldsymbol{\alpha}| \leq \nu_*} \|D^{(\boldsymbol{\alpha})}\Psi(\cdot)\|_{L^\infty}.$$

Being the minimum over a finite set, to prove $c_\nu > 0$ corresponds to proving that all $\|D^{(\boldsymbol{\alpha})}\Psi(\cdot)\|_{L^\infty}$ are not zero. To this aim, note that $\|D^{(\boldsymbol{\alpha})}\Psi(\cdot)\|_{L^\infty} = 0$ would imply that $D^{(\alpha)}\Psi(\cdot) = 0$, which is false as $\Psi$ is not a polynomial function. $\square$

At this point, we define a squeezed version of $\Psi$, which we call:

$$\rho < 1: \qquad \Psi_\rho(x) := \Psi(x/\rho). \tag{53}$$

About this function, we prove a crucial lemma that we are going to use in the proof of the lower bound.

**Lemma 3.** *Fix $\nu > 0$ and let $c_\nu$ be defined as in Lemma 2. For any $|\boldsymbol{\alpha}| \leq \nu_*$, we have*

$$\|D^{(\boldsymbol{\alpha})}\Psi_\rho(\cdot)\|_{L^\infty} \geq c_\nu \rho^{-|\boldsymbol{\alpha}|}$$

*Proof.* By the properties of the derivative, we have

$$\forall \boldsymbol{\alpha} \qquad D^{(\boldsymbol{\alpha})}\Psi_\rho(\cdot) = \rho^{-|\alpha|} D^{(\boldsymbol{\alpha})}\Psi(\cdot).$$

This proves that

$$\|D^{(\boldsymbol{\alpha})}\Psi_\rho(\cdot)\|_{L^\infty} = \rho^{-|\boldsymbol{\alpha}|}\|D^{(\boldsymbol{\alpha})}\Psi(\cdot)\|_{L^\infty} \geq c_\nu \rho^{-|\boldsymbol{\alpha}|}.$$

$\square$

This result can be completed with the following upper bound.

**Lemma 4.** $\|\Psi_\rho\|_{\mathcal{C}^\nu} \leq \|\Psi\|_{\mathcal{C}^\nu}\rho^{-\nu}.$

*Proof.* We first prove that, for every $|\boldsymbol{\alpha}| < \nu$, we have the correct bound on the infinity norm.

$$\|D^{(\boldsymbol{\alpha})}\Psi_\rho\|_{L^\infty} = \rho^{-|\boldsymbol{\alpha}|}\|D^{(\boldsymbol{\alpha})}\Psi\|_{L^\infty} \leq \rho^{-\nu}\|\Psi\|_{\mathcal{C}^\nu}.$$

Next, we need to bound $L_\nu(\Psi_\rho)$. By definition,

$$
\begin{aligned}
L_\nu(\Psi_\rho) &= \max_{|\boldsymbol{\alpha}|=\nu_*} \sup_{x \neq y} \frac{|D^{(\boldsymbol{\alpha})}\Psi_\rho(x) - D^{(\boldsymbol{\alpha})}\Psi_\rho(y)|}{|x-y|^{\nu-\nu_*}} \\
&= \max_{|\boldsymbol{\alpha}|=\nu_*} \sup_{x \neq y} \frac{|\rho^{-\nu_*}D^{(\boldsymbol{\alpha})}\Psi(x/\rho) - \rho^{-\nu_*}D^{(\boldsymbol{\alpha})}\Psi(y/\rho)|}{|x-y|^{\nu-\nu_*}} \\
&= \rho^{-\nu_*} \max_{|\boldsymbol{\alpha}|=\nu_*} \sup_{x \neq y} \frac{|D^{(\boldsymbol{\alpha})}\Psi(x/\rho) - D^{(\boldsymbol{\alpha})}\Psi(y/\rho)|}{|x-y|^{\nu-\nu_*}} \\
&= \rho^{-\nu_*}\rho^{-\nu+\nu_*} \max_{|\boldsymbol{\alpha}|=\nu_*} \sup_{x \neq y} \frac{|D^{(\boldsymbol{\alpha})}\Psi(x/\rho) - D^{(\boldsymbol{\alpha})}\Psi(y/\rho)|}{|x/\rho-y/\rho|^{\nu-\nu_*}} \\
&= \rho^{-\nu} \sup_{x \neq y} \frac{|D^{(\boldsymbol{\alpha})}\Psi(x/\rho) - D^{(\boldsymbol{\alpha})}\Psi(y/\rho)|}{|x/\rho-y/\rho|^{\nu-\nu_*}} \\
&= \rho^{-\nu} L_\nu(\Psi).
\end{aligned}
$$

This ends the proof. $\qquad\square$

With all these lemmas, we can proceed to the actual proof of the lower bound.

### F.2 Proof of the lower bound

**Theorem 10.** *Any algorithm outputting an estimation $D^{(\boldsymbol{\alpha})}\widehat{f}_n(\cdot)$ for $D^{(\boldsymbol{\alpha})}f(\cdot)$ suffers an error lower-bounded by*

$$
\|D^{(\boldsymbol{\alpha})}f(\cdot) - D^{(\boldsymbol{\alpha})}\widehat{f}_n(\cdot)\| \geq \Omega\left(n^{-\frac{\nu-|\boldsymbol{\alpha}|}{2\nu+d}}\psi_0^{\frac{2|\boldsymbol{\alpha}|+d}{2\nu+d}}\sigma^{\frac{2\nu-2|\boldsymbol{\alpha}|}{2\nu+d}}\right),
$$

*w.p. at least $1/4$, on a problem instance satisfying assumptions 1 and 2 with $\|f(\cdot)\|_{\mathcal{C}^\nu} \leq \psi_0$.*

*Proof.* Let $K \in \mathbb{N}$. Let us divide $[-1,1]^d$ into $K^d$ hypercubes $\{Q_{\boldsymbol{\ell}}\}_{\boldsymbol{\ell}=(1,1,\dots1)}^{(K,K,\dots K)}$, each of side $\rho := 2/K$ and center $q_{\boldsymbol{\ell}}$ respectively. By definition of the sampling process, there is $\boldsymbol{\ell}_\star$ such that

$$
\sum_{i=1}^n \mathbf{1}(x_i \in Q_{\boldsymbol{\ell}_\star}) \leq n/K^d.
$$

Let us define two problem instances, both affected by a noise $\mathcal{N}(0,\sigma^2)$:

$$
f_1(\cdot) = 0 \qquad f_2(\cdot) = \begin{cases} 0 & Q_{\boldsymbol{\ell}_\star}^c \\ \psi_0\|\Psi\|_{\mathcal{C}^\nu}^{-1}\rho^\nu \Psi_\rho(\boldsymbol{x}-q_{\boldsymbol{\ell}_\star}) & Q_{\boldsymbol{\ell}_\star} \end{cases}, \tag{54}
$$

We have to verify that both functions are in $\mathcal{C}_p^\nu([-1,1]^d)$. For the first function, the result is trivial, while for the second one, we have that i) no derivative can be discontinuous on $\dot{Q}_{\boldsymbol{\ell}_\star}$ or $\dot{Q}_{\boldsymbol{\ell}_\star}^c$, as both $\Psi_\rho(\cdot)$ and $0$ are infinitely times differentiable ii) On the boundaries between the hypercubes all function and all their derivatives are identically zero, as it follows from the definition of $\Psi$.

We can now evaluate the norm of both functions. $\|f_1(\cdot)\|_{L^\infty} = \|f_1(\cdot)\|_{\mathcal{C}^\nu} = 0$, while for the second function we have, due to Lemma 4,

$$
\|f_2(\cdot)\|_{L^\infty} = \psi_0\|\Psi\|_{\mathcal{C}^\nu}^{-1}\rho^\nu, \qquad \|\Psi_\rho\|_{\mathcal{C}^\nu} \leq \psi_0\|\Psi\|_{\mathcal{C}^\nu}^{-1}\rho^\nu\|\Psi\|_{\mathcal{C}^\nu}\rho^{-\nu} = \psi_0. \tag{55}
$$

With this result fixed, we are able to measure the KL divergence between the two distributions of the data $y_i$ under the true function $f_1(\cdot)$ and $f_2(\cdot)$, respectively. Let us call $P_1(y_1, \dots y_i, \dots y_n)$ the distribution of the samples if the true function is $f_1(\cdot)$. We have

$$P_1(y_1, \ldots y_i, \ldots y_n) = \prod_{i=1}^{n} \mathcal{N}(y_i, \sigma^2).$$

In the same way, for what concerns $P_2$, we have

$$P_2(y_1, \ldots y_i, \ldots y_n) = \prod_{i=1}^{n} \mathcal{N}(y_i - f_2(x_i), \sigma^2).$$

The KL divergence between the two measures corresponds, by definition, to

$$KL(P_1, P_2) = \int_{\mathbb{R}^n} \log \left( \prod_{i=1}^{n} \frac{\mathcal{N}(y_i, \sigma^2)}{\mathcal{N}(y_i - f_2(x_i), \sigma^2)} \right) \prod_{i=1}^{n} \mathcal{N}(y_i, \sigma^2) dy_i \tag{56}$$

$$\leq \int_{\mathbb{R}^n} \log \left( \prod_{i=1}^{n/K^d} \frac{\mathcal{N}(y_i, \sigma^2)}{\mathcal{N}(y_i - f_2(x_i), \sigma^2)} \right) \prod_{i=1}^{n} \mathcal{N}(y_i, \sigma^2) dy_i. \tag{57}$$

Where the second passage follows from the fact that $f_2(x_i) = 0$ for all but at most $n/K^d$ samples, which we assume without loss of generality to be the first. The sum can now be written as

$$KL(P_1, P_2) \leq \int_{\mathbb{R}^n} \sum_{i=1}^{n/K^d} \log \left( \frac{\mathcal{N}(y_i, \sigma^2)}{\mathcal{N}(y_i - f_2(x_i), \sigma^2)} \right) \prod_{i=1}^{n} \mathcal{N}(y_i, \sigma^2) dy_i \tag{58}$$

$$= \sum_{i=1}^{n/K^d} \int_{\mathbb{R}^n} \log \left( \frac{\mathcal{N}(y_i, \sigma^2)}{\mathcal{N}(y_i - f_2(x_i), \sigma^2)} \right) \mathcal{N}(y_i, \sigma^2) dy_i, \tag{59}$$

indeed, all other variables integrate to one. At this point, using lemma 1, we have

$$KL(P_1, P_2) \leq \sum_{i=1}^{n/K^d} \int_{\mathbb{R}^n} \log \left( \frac{\mathcal{N}(y_i, \sigma^2)}{\mathcal{N}(y_i - f_2(x_i), \sigma^2)} \right) \mathcal{N}(y_i, \sigma^2) dy_i \tag{60}$$

$$= \sum_{i=1}^{n/K^d} \frac{f_2(x_i)^2}{2\sigma^2} \leq \frac{n \psi_0^2 \rho^{2\nu}}{2 K^d \|\Psi\|_{\mathcal{C}^\nu}^2 \sigma^2}. \tag{61}$$

From this, it follows that any algorithm $\mathcal{A}$ that classifies between $f_1$ and $f_2$ has an error probability at least (Theorem 2.2 Tsybakov (2009))

$$p_{1,e} \geq \frac{1 - \sqrt{(n/K^d)\|\Psi\|_{\mathcal{C}^\nu}^{-2} \psi_0^2 \rho^{2\nu}/(4\sigma^2)}}{2}.$$

Remembering that in the former we have $\rho = 2/K$, the previous writes as

$$p_{1,e} \geq \frac{1 - (2\sigma \|\Psi\|_{\mathcal{C}^\nu})^{-1} \psi_0 \sqrt{n/K^{d+2\nu}}}{2}.$$

Therefore, the previous inequality lets us find the minimum $K$ such that the probability of making is bounded from below by $1/4$:

$$K \geq \left( \frac{\psi_0^2 n}{4\sigma^2 \|\Psi\|_{\mathcal{C}^\nu}^2} \right)^{\frac{1}{d+2\nu}} \implies p_{1,e} \geq 1/4. \tag{62}$$

Now, following a standard argument for lower bounds, note that if no *classifier* $\mathcal{A}$ can distinguish the two functions $f_1, f_2$ with a given probability, no regression algorithm producing $\widehat{f}_n$ can achieve error less than $\|f_1(\cdot) - f_2(\cdot)\|/2$ with the same probability in *any* norm $\|\cdot\|$. Otherwise, a trivial classifier $\mathcal{A}$ that outputs the function minimizing, between $f_1, f_2$, the distance (w.r.t $\|\cdot\|$) from $\widehat{f}_n$ would violate this condition. In our specific case, we are interested in the infinity norm difference between order $\boldsymbol{\alpha}$ derivatives, which is

$$\|D^{(\boldsymbol{\alpha})} f_1(\cdot) - D^{(\boldsymbol{\alpha})} f_2(\cdot)\|_{L^\infty} = \psi_0 \|D^{(\boldsymbol{\alpha})} \Psi_\rho(\cdot)\|_{L^\infty}$$

$$\geq \psi_0 \|\Psi\|_{\mathcal{C}^\nu}^{-1} \rho^\nu c_\nu \rho^{-|\boldsymbol{\alpha}|} = \frac{\psi_0 c_\nu}{\|\Psi\|_{\mathcal{C}^\nu}} \rho^{\nu - |\boldsymbol{\alpha}|},$$

where the last passage comes from lemma 3. Replacing $\rho = 2/K$ we get

$$\|D^{(\boldsymbol{\alpha})} f_1(\cdot) - D^{(\boldsymbol{\alpha})} f_2(\cdot)\|_{L^\infty} \geq \frac{\psi_0 c_\nu}{\|\Psi\|_{\mathcal{C}^\nu}} (K/2)^{|\boldsymbol{\alpha}| - \nu}.$$

Replacing the lower bound for $K$ that we got in equation (62) proves that, w.p. $1/4$, the error in estimating any derivative $\boldsymbol{\alpha}$ is

$$\|D^{(\boldsymbol{\alpha})} f(\cdot) - D^{(\boldsymbol{\alpha})} \widehat{f}_n(\cdot)\| \geq \frac{\psi_0^{\frac{d+2|\boldsymbol{\alpha}|}{d+2\nu}} c_\nu}{\|\Psi\|_{\mathcal{C}^\nu}} 2^{\nu - |\boldsymbol{\alpha}|} \left( \frac{n}{4\sigma^2 \|\Psi\|_{\mathcal{C}^\nu}^2} \right)^{\frac{|\boldsymbol{\alpha}| - \nu}{d+2\nu}}.$$

This ends the proof.

$\square$

### F.3 Space complexity lower bound

**Theorem 11.** *Any algorithm with optimal statistical complexity for a regression problem satisfying assumptions 1 and 2 must have a space complexity in the prediction of at least $\Omega\left(n^{\frac{d}{2\nu+d}}\right)$.*

*Proof.* The proof strongly relies on lemma 5: due to this result, there are $J = \Omega(2^{\varepsilon^{-d/\nu}})$ functions in $\mathcal{C}_p^\nu([-1,1]^d)$ such that $\|f_j\|_{\mathcal{C}^\nu} \leq 1$ and

$$\forall j \neq j' \qquad \|f_j - f_{j'}\|_{L^\infty} \geq \varepsilon.$$

Any algorithm that attains error $J$ must at least be able to distinguish these $J$ instances. Therefore, it has to occupy a number of bits given by $\log_2(J) = \Omega(\varepsilon^{-d/\nu})$. As we have seen in the main paper (theorem 10), the order-optimal guarantee on the statistical complexity links the error $\varepsilon$ to the number of samples in the following way

$$\varepsilon = \Omega\left(n^{-\frac{\nu}{2\nu+d}}\right),$$

meaning that the total number of bits is given by $\Omega\left(n^{-\frac{\nu}{2\nu+d} \frac{-d}{\nu}}\right) = \Omega\left(n^{\frac{d}{2\nu+d}}\right)$. This ends the proof. $\square$

**Lemma 5.** *Fix $\varepsilon > 0$. There is family of functions $\{f_j\}_{j=1}^J \subset \mathcal{C}_p^\nu([-1,1]^d)$ such that $\|f_j\|_{\mathcal{C}^\nu} \leq 1$ and*

$$\forall j \neq j' \qquad \|f_j(\cdot) - f_{j'}(\cdot)\|_{L^\infty} \geq \varepsilon, \qquad J = \Omega\left(2^{\varepsilon^{-d/\nu}}\right)$$

*Proof.* For this proof, we are going to use the notation of the previous part of this section.

Let us divide $[-1,1]^d$ into $K^d$ hypercubes $\{Q_\ell\}_{\ell=1}^{K^d}$, each of side $\rho := 2/K$ and center $q_\ell$ respectively. Define, for every $j = [2^{K^d}]$ the following function:

$$f_j(\cdot) = \sum_{\ell=1}^{K^d} \text{bin}(j,\ell) 1_{Q_\ell}(\cdot) \|\Psi\|_{\mathcal{C}^\nu}^{-1} \rho^\nu \Psi_\rho(\cdot - q_\ell), \tag{63}$$

where $\text{bin}(j,\ell)$ indicates that the binary digit for $j$ (which has at most $K^d$ digits) in position $\ell$. In section F.1 and specifically lemma 4, we have already proved that

$$1_{Q_\ell}(\cdot) \|\Psi\|_{\mathcal{C}^\nu}^{-1} \rho^\nu \Psi_\rho(\cdot - q_\ell) \in \mathcal{C}_p^\nu([-1,1]^d), \tag{64}$$

with $\|\cdot\|_{\mathcal{C}^\nu}$ norm bounded by one. Since, in equation (63), we are summing over functions of this form with disjoint supports, the same result applies here, showing that $\|f_j\|_{\mathcal{C}^\nu} \le 1$. Now, take two $f_j, f_{j'}$ for $j \ne j'$ and measure their infinity norm difference. As $j \ne j'$, there must be $\ell$ such that $\text{bin}(j,\ell_0) \ne \text{bin}(j',\ell_0)$.

$$\begin{aligned}
\|f_j(\cdot) - f_{j'}(\cdot)\|_{L^\infty} &\ge \|f_j(\cdot) - f_{j'}(\cdot)\|_{L^\infty(Q_{\ell_0})} \\
&= \|\Psi\|_{\mathcal{C}^\nu}^{-1} \|1_{Q_\ell}(\cdot)\rho^\nu \Psi_\rho(\cdot - q_\ell)\|_{L^\infty} \\
&= \|\Psi\|_{\mathcal{C}^\nu}^{-1} \rho^\nu \|1_{Q_\ell}(\cdot)\Psi_\rho(\cdot - q_\ell)\|_{L^\infty} = \|\Psi\|_{\mathcal{C}^\nu}^{-1} \rho^\nu
\end{aligned}$$

(the last passage is due to the fact that $\|\Psi_\rho(\cdot - q_\ell)\|_{L^\infty} = 1$, as we can simply see from its definition). Therefore, for $\|\Psi\|_{\mathcal{C}^\nu}^{-1} \rho^\nu \ge \varepsilon$, the condition in the statement is satisfied. In turn, this implies,

$$\rho^\nu \ge \|\Psi\|_{\mathcal{C}^\nu} \varepsilon \implies \rho \ge (\|\Psi\|_{\mathcal{C}^\nu} \varepsilon)^{1/\nu}.$$

Replacing $\rho = 2/K$, this condition corresponds to $K \le 2(\|\Psi\|_{\mathcal{C}^\nu} \varepsilon)^{1/\nu}$; which is satisfied by $K = \lfloor 2(\|\Psi\|_{\mathcal{C}^\nu} \varepsilon)^{-1/\nu} \rfloor$, i.e.

$$J = 2^{K^d} = 2^{\lfloor 2(\|\Psi\|_{\mathcal{C}^\nu} \varepsilon)^{-1/\nu} \rfloor^d} = \Omega\left(2^{\varepsilon^{-d/\nu}}\right).$$

$\square$

# G   Extension to non-periodic functions

In this section, we are going to show how it is possible to generalize to non-periodic functions if we allow the agent to take samples outside of $\mathcal{X}$. Apparently, the agent has no advantage in querying a point that does not belong to this set, as the $L^\infty$ error is only evaluated on $\mathcal{X}$. Still, we can leverage this to avoid the periodicity assumption. By simplicity, we will fix $\mathcal{X} = [-1,1]^d$ as in the main paper and $\mathcal{Y} = [-3,3]^d$ (the set where we allow queries).

Recall that from the results in section F.1 for $d = 1$, there is $\Psi : \mathbb{R} \to [0,1]$ that is infinitely many times differentiable, $\Psi(0) = 1$ and $\psi(x) = 1$ if $x > 1$ or $x < -1$.

Let us define the following functions $\mathbb{R}^d \to [0,1]$ and $\mathbb{R} \to [0,1]$.

$$H(x) := \prod_{j=1}^d h(x_j) \qquad h(x) := \frac{\int_{-2+x}^{2+x} \Psi(y)dy}{\int_{-1}^1 \Psi(y)dy}. \tag{65}$$

By construction, the following properties hold:

1.  $h$ is in $\mathcal{C}^\infty(\mathbb{R})$, as it can be seen as a convolution between $\Psi$, that is $\mathcal{C}^\infty(\mathbb{R})$ and $1_{[-2,2]}(x)$.

2. For $x \in [-1, 1]$ we have

$$h(x) = \frac{\int_{-2+x}^{2+x} \Psi(y) dy}{\int_{-1}^{1} \Psi(y) dy} = \frac{\int_{-1}^{1} \Psi(y) dy}{\int_{-1}^{1} \Psi(y) dy} = 1,$$

since $\Psi$ is compactly supported on $[-1, 1]$.

3. For $x > 3$:

$$h(x) = \frac{\int_{-2+x}^{2+x} \Psi(y) dy}{\int_{-1}^{1} \Psi(y) dy} \leq \frac{\int_{1}^{2+x} \Psi(y) dy}{\int_{-1}^{1} \Psi(y) dy} = 0,$$

and the same holds true for $x < -3$.

From these three properties, along with the fact that the product of smooth functions is smooth, we have the following lemma.

**Lemma 6.** *The function $H$ defined in equation (65) belongs to $\mathcal{C}^\infty(\mathbb{R}^d)$. Moreover,*

$$H(x) : \begin{cases} = 1 & x \in \mathcal{X} \\ \in [0, 1] & x \in \mathcal{Y} \setminus \mathcal{X} \, . \\ = 0 & x \in \mathcal{Y}^c \end{cases}$$

At this point, we come to the key observation that allows us to generalize our method to non-periodic functions.

Assume that we want to estimate $f \in \mathcal{C}^\nu(\mathbb{R}^d)$ on $\mathcal{X}$, and we are allowed to take samples from $\mathcal{Y}$. We cannot directly run algorithm 1 on $f$, which is not periodic. However, we can run it on

$$f^\star(x) := f(x)H(x),$$

that *is* periodic on $\mathcal{Y}$, as all its derivatives at the boundary of $\mathcal{Y}$ are identically zero. In this way, algorithm 1 ensures that we can uniformly approximate $f(x)H(x)$ for $x \in \mathcal{Y}$ and, as $H$ is identically one on $\mathcal{X}$, this also provides a uniform bound for $f$ on $\mathcal{X}$.

This generalized algorithm works as in 2. As $H(\cdot) \in C^\infty(\mathbb{R}^d)$, the function $f^\star$ is as smooth as $f$ on $\mathcal{Y}$. Moreover, as $|H(\cdot)| \leq 1$, multiplying by this number does not change the sub-Gaussian constant of the samples. Therefore, the sample complexity guarantee for algorithm 2 follows directly from that of DUPA.

**Restricting all queries to be inside the domain.** The previous procedure requires the agent to query values $x$ that are outside of $[-1, 1]^d$, the domain of $f$. While by simplicity, we have shown a procedure in which the generalized samples are queried in $[-3, 3]^d$, a straightforward adaptation shows that $[-1 - \xi, 1 + \xi]^d$, for some $\xi > 0$, is sufficient. Still, in some real situations, the agent is constrained to access the values of $f$ only in the original domain, without any possibility of trespassing.

In this case, to maintain the validity of the former analysis, it is necessary to apply the extension theorem to estimate $f$ on $[-1 - \xi, 1 + \xi]^d$ using only valid samples. In Hestenes (1941), it was proved that a function $f \in \mathcal{C}^\nu([a, b]^d)$ can be extended to a larger domain by defining the values outside the boundary as a linear combination of the values at specific mirrored points inside the domain. This procedure maintains the same number of continuous derivatives. For example, to extend a function $f$ from $[0, 1]$ to $[-1, 1]$, the extension for $x < 0$ is defined as:

$$\tilde{f}(x) = \sum_{i=1}^{\nu+1} c_i f(-\lambda_i x)$$

where $\lambda_i > 0$ ensures that $-\lambda_i x \in [0, 1]$. By leveraging this procedure, the agent can compute the required out-of-domain values by querying only valid internal samples, effectively overcoming the boundary constraint without trespassing.

---

**Algorithm 2** Generalized DUPA

---

**Require:** Query space $\mathcal{Y}$, max number of samples $n$, feature map length $N$, smoothness order $\nu > 0$
**Ensure:** Estimated function $\widehat{f}_n$ and its derivatives $\widehat{f}_n^{(1)}, \ldots, \widehat{f}_n^{(\nu_\star)}$
  1: Define $H(\cdot)$ as in (65)
  2: (... next steps go as DUPA)
  3: **for all** $x \in \mathrm{supp}(\rho)$ **do**
  4:     **for** $j = 1$ to $\lceil n_{\mathrm{tot}} \cdot \rho(x) \rceil$ **do**
  5:       (...)
  6:       Query $y_i^+$ at $x + \eta_+$ and multiply the result by $H(x + \eta_-)$ (former 13)
  7:       Query $y_i^-$ at $x + \eta_-$ and multiply the result by $H(x + \eta_-)$ (former 14)
  8:       (...)
  9:     **end for**
10: **end for**
11: (...)
12: **return** $\widehat{f}_n(\cdot) = \phi_N(\cdot)^\top \widehat{\theta}_n$ and its derivatives restricted to $\mathcal{X}$.

---

# H  Numerical Simulations: Extensions and Ablation Studies

In this section, we provide additional simulations to further corroborate our theoretical findings. Specifically, the objectives of this evaluation are threefold:

1. **Multivariate Extension:** While the experiments in the main text focus on a univariate regression problem (i.e., curve fitting on the interval $[-1, 1]$), this section evaluates the proposed method on a two-dimensional benchmark to assess its scalability to higher dimensions.

2. **Derivative Approximation:** In the primary analysis, performance was measured solely based on the approximation of the target function. However, approximating derivatives is inherently ill-posed, particularly with empirical data, where derivatives may be ill-defined or unbounded. Here, we utilize a synthetic, differentiable benchmark to quantify the approximation error with respect to the first-order derivative.

3. **Ablation Study:** While previous simulations compared our approach against state-of-the-art non-parametric regression baselines to demonstrate advantages in performance and computational complexity, we now focus on an internal comparison. We evaluate a variant of our algorithm using the standard Dirichlet kernel (figure 1) instead of the more sophisticated De La Vallée Poussin kernel. This comparison serves to justify the necessity of the algorithmic complexity introduced by the latter.

The bivariate case introduces significant challenges compared to the univariate setting, primarily due to the exponential growth of resource requirements. Discretizing the domain with equivalent precision requires a quadratic increase in the number of samples. Similarly, a Fourier map of degree $N$ results in a feature space dimension that scales quadratically, significantly increasing the computational overhead.

**Benchmark Specification**    To evaluate the performance in this setting, we employ the following target function $f : [-1, 1]^2 \to \mathbb{R}$:

$$f(x_1, x_2) = \exp(-5x_1^2 - 5x_2^2). \tag{66}$$

This function is smooth and infinitely differentiable across the entire domain, facilitating the exact computation and subsequent approximation of its derivatives. Due to the radial symmetry of $f$, we focus our analysis on the approximation of the function itself and its partial derivative with respect to the first variable:

$$\frac{\partial f}{\partial x_1}(x_1, x_2) = -10x_1 \exp(-5x_1^2 - 5x_2^2).$$

In the next section, we present the results gotten.

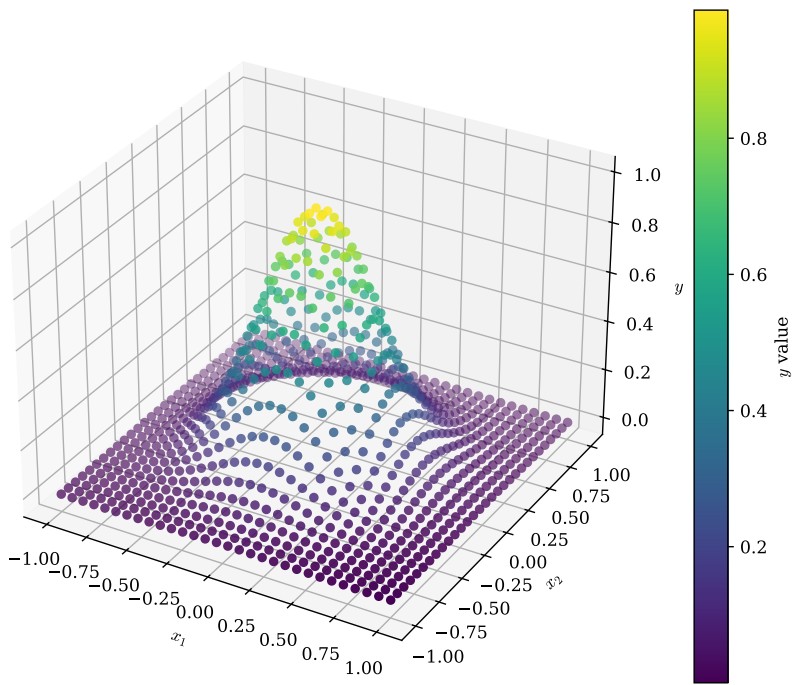

Figure 4: Benchmark function $f$ from equation (66). Axis $x_1$ and $x_2$ represent the two input coordinates. Height and color represent $f(x)$.

.

## H.1   Experiment results

We evaluated the performance of the DUPA framework (Algorithm 1) on the bivariate regression task under two distinct configurations. Specifically, we compared a baseline implementation using the standard Dirichlet kernel with the proposed version using the De La Vallée Poussin (DVP) kernel. Initial qualitative results are illustrated in Figure 5, where we report the estimators obtained using $n = 10^4$ samples corrupted by additive Gaussian noise ($\sigma = 0.05$). We set the Fourier feature map degree to $N = 6$, resulting in a total of $N_d = \binom{2N+d}{d} = \binom{14}{2} = 91$ parameters. This choice ensures an overdetermined system ($n \gg N_d$), which is appropriate for evaluating the approximation capabilities of the kernels. As shown in the plots, the DVP variant demonstrates clear superiority in reconstruction fidelity. While the DVP estimator accurately recovers the geometry of the target function $f$, the Dirichlet-based reconstruction exhibits significant oscillatory artifacts across the domain, a manifestation of the well-known Gibbs phenomenon and the lack of localization of the Dirichlet kernel. To provide a rigorous quantitative validation of these observations, we conducted extensive simulations across varying dataset sizes and random seeds. These results are summarized in Table 2.

Table 2 summarizes the results of the regression experiments. Specifically, the two algorithms were fitted using dataset sizes $n$ ranging from 1000 to 50000, as shown in the first row. For each value of $n$, the experiment was repeated across five random seeds; the table reports the mean $\pm$ standard deviation for two error metrics: the $L^2$ norm, measuring the mean square error, and the $L^\infty$ norm, which is the primary focus of this study. As expected, the results indicate that the uniform error is consistently higher than the mean square error. Furthermore, estimating the derivative of the function proves more challenging than estimating the function itself, and all errors decrease as the sample size $n$ increases. Notably, the performance of PDUA with the DVP kernel significantly outperforms the Dirichlet kernel across all experiments. Regarding function approximation, the DVP kernel with only 1000 data points achieves better results than

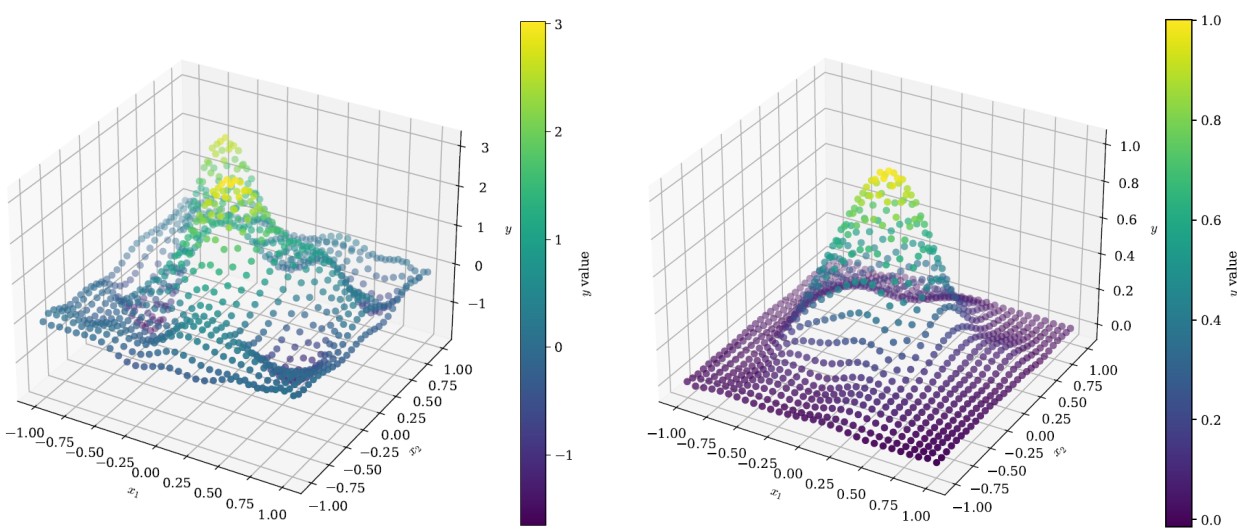

Figure 5: Comparison between the DUPA algorithm (right) and its naive version, which uses the Dirichlet kernel in place of the De La Vallèe Poussin one (left). The plots show approximations of $f$ using noisy samples.

Table 2: Comparison of the approximation errors for DUPA implemented with the Dirichlet and the De la Vallèe Poussin (DVP) kernels. In the columns, we let the sample size vary; in the rows, we alternate between the two methods as well as between the estimation of the function and its derivative, and the type of metric.

| Target | Method | Metric | $10^3$ | $2 \times 10^3$ | $5 \times 10^3$ | $10^4$ | $2 \times 10^4$ | $5 \times 10^4$ |
|---|---|---|---|---|---|---|---|---|
| $f$ | Dirichlet | $L^2$ | $1.49 \pm 0.28$ | $1.02 \pm 0.06$ | $0.72 \pm 0.15$ | $0.52 \pm 0.11$ | $0.4 \pm 0.04$ | $0.39 \pm 0.02$ |
| | | $L^\infty$ | $5.05 \pm 1.0$ | $3.47 \pm 0.25$ | $2.5 \pm 0.75$ | $1.94 \pm 0.37$ | $1.46 \pm 0.24$ | $1.61 \pm 0.07$ |
| | **DVP** | $L^2$ | $0.08 \pm 0.03$ | $0.06 \pm 0.01$ | $0.04 \pm 0.01$ | $0.03 \pm 0.01$ | $0.02 \pm 0.01$ | $0.01 \pm 0.0$ |
| | | $L^\infty$ | $0.32 \pm 0.15$ | $0.23 \pm 0.07$ | $0.15 \pm 0.06$ | $0.12 \pm 0.04$ | $0.09 \pm 0.03$ | $0.05 \pm 0.01$ |
| $\frac{\partial f}{\partial x_1}$ | Dirichlet | $L^2$ | $8.22 \pm 1.9$ | $5.48 \pm 0.86$ | $4.01 \pm 0.74$ | $2.69 \pm 0.7$ | $1.85 \pm 0.18$ | $1.5 \pm 0.09$ |
| | | $L^\infty$ | $30.06 \pm 7.69$ | $17.15 \pm 3.5$ | $12.06 \pm 2.76$ | $10.28 \pm 3.71$ | $6.32 \pm 0.54$ | $4.98 \pm 0.7$ |
| | **DVP** | $L^2$ | $0.44 \pm 0.15$ | $0.32 \pm 0.06$ | $0.24 \pm 0.08$ | $0.19 \pm 0.02$ | $0.13 \pm 0.03$ | $0.09 \pm 0.02$ |
| | | $L^\infty$ | $1.51 \pm 0.56$ | $0.99 \pm 0.23$ | $0.81 \pm 0.37$ | $0.62 \pm 0.12$ | $0.41 \pm 0.09$ | $0.3 \pm 0.06$ |

the Dirichlet kernel with 50000 points, maintaining a substantial margin (the error in the latter case is approximately five times higher in both the uniform and mean square norms). A similar pattern emerges for derivative estimation, where the DVP kernel remains dominant even with a much smaller dataset. This ranking remains robust even when accounting for experimental uncertainty. Interestingly, the gap between the two methods observed in these experiments is far more pronounced than theoretical predictions suggest. While the theoretical difference is characterized only by logarithmic factors, the empirical results show a stark divergence. This suggests that the DVP kernel provides even more robust performance in non-asymptotic regimes than the current theory accounts for.

# I Fundamental Results

**Theorem 16** (Vector-valued Bernstein's inequality)**.** *Let $\{x_i, y_i\}_{i=1}^n$ be a dataset of points where $x_i$ is deterministic and for some feature map*

$$y_i = \phi(x_i)^\top \theta + \xi_i, \qquad \theta \in \mathbb{R}^d$$

*where $\{\xi_i\}_{i=1}^n$ is a family of independent r.v. with variance $\gamma^2$ and bounded by $B$ almost surely. Let $\{\phi(x_i)\}_{i=1}^n \subset \mathcal{V} \subset \mathbb{R}^d$ and define*

$$V_\star := \sup_{v \in \mathcal{V}} \|v\|_{\Sigma_n^{-1}}^2.$$

*With probability at least $1 - \delta$,*

$$|(\widehat{\theta}_n - \theta)^\top v| \leq \sqrt{2V_\star \gamma^2 \log(1/\delta)} + \frac{2V_\star B \log(1/\delta)}{3}.$$

*In the special case of a quasi-optimal design, where $V_\star = \sqrt{2d/n}$, the former gives*

$$\forall v \in \mathcal{V} \qquad |(\widehat{\theta}_n - \theta)^\top v| \leq \sqrt{\frac{4d\gamma^2 \log(|\mathcal{V}|/\delta)}{n}} + \frac{4dB \log(|\mathcal{V}|/\delta)}{3n}.$$

*Proof.* By definition,

$$\widehat{\theta}_n = \Sigma_n^{-1} \sum_{i=1}^n \phi(x_i) y_i = \theta + \Sigma_n^{-1} \sum_{i=1}^n \phi(x_i)\xi_i,$$

where $\Sigma_n := \sum_{i=1}^n \phi(x_i)\phi(x_i)^\top$. Take any $v \in \mathcal{V}$. It follows that

$$(\widehat{\theta}_n - \theta)^\top v = v^\top \Sigma_n^{-1} \sum_{i=1}^n \phi(x_i)\xi_i.$$

Being the variables independent, the variance of the whole random variable writes as

$$\begin{aligned}
\mathrm{Var}\left[(\widehat{\theta}_n - \theta)^\top v\right] &= \gamma^2 \sum_{i=1}^n (v^\top \Sigma_n^{-1}\phi(x_i))^2 \\
&= \gamma^2 \sum_{i=1}^n v^\top \Sigma_n^{-1}\phi(x_i)\phi(x_i)^\top \Sigma_n^{-1}v \\
&= \gamma^2 v^\top \left(\sum_{i=1}^n \Sigma_n^{-1}\phi(x_i)\phi(x_i)^\top\right) \Sigma_n^{-1}v \\
&= \gamma^2 \|v\|_{\Sigma_n^{-1}}^2 \leq \gamma^2 V_\star.
\end{aligned}$$

At the same time, the sum can be written as

$$v^\top \Sigma_n^{-1} \sum_{i=1}^n \phi(x_i)\xi_i = \frac{1}{n}\sum_{i=1}^n v^\top n\Sigma_n^{-1}\phi(x_i)\xi_i,$$

where the term inside the sum does not exceed $\|v\|_{n\Sigma_n^{-1}}\|\phi(x_i)\|_{n\Sigma_n^{-1}}B \leq nV_\star B$. Therefore, Bernstein's inequality (Boucheron et al., 2003) ensures that, with probability at least $1 - \delta$,

$$|(\widehat{\theta}_n - \theta)^\top v| \leq \sqrt{2V_\star \gamma^2 \log(1/\delta)} + \frac{2V_\star B \log(1/\delta)}{3}.$$

Making a union bound on the $k$ values for $v$ gives the thesis. $\qquad\square$

**Theorem 17.** *Let $T_N \in \mathbb{T}_N$. Then, $\|T_N^{(1)}(\cdot)\|_{L^\infty} \leq 4\pi N \|T_N(\cdot)\|_{L^\infty}$.*

*Proof.* This result follows from the classical Bernstein inequality for algebraic polynomials. Indeed, a trigonometric polynomial

$$T_N(x) = \sum_{|k| \leq N} c_k e^{2\pi i k x}$$

can be written as

$$T_N(x) = e^{-2\pi i N x} P(e^{2\pi i x}),$$

where $P$ is a polynomial of degree at most $2N$. Since $|e^{2\pi i x}| = 1$, we may apply Bernstein's inequality for complex polynomials,

$$\sup_{|z|=1} |P'(z)| \leq N \sup_{|z|=1} |P(z)|,$$

which yields the desired bound. $\qquad\square$

An analog theorem holds in dimension $d > 1$.

**Theorem 18.** *Let $T_N \in \mathbb{T}_{d,N}$. Then, $\|\|\nabla T_N(\cdot)\|_\infty\|_{L^\infty} \leq 4\pi N \|T_N(\cdot)\|_{L^\infty}$.*

*Proof.* By the definition of gradient,

$$\nabla T_N(\cdot) = [\partial_1 T_N(\cdot), \dots \partial_d T_N(\cdot)],$$

where $\partial_j = D^{(0, \dots \underbrace{1}_{j}, \dots 0)}$ correspond to the partial derivative w.r.t. index $j$. Note that, considering only the $j-$th variable, $T_N(\cdot)$ is still a trigonometric polynomial with a degree at most $N$. Therefore, theorem 17 ensures,

$$\|\partial_j T_N(\cdot)\|_{L^\infty} \leq 4\pi N \|T_N(\cdot)\|_{L^\infty}.$$

Using this property, we have

$$\|\|\nabla T_N(\cdot)\|_\infty\|_{L^\infty} = \|\max_{j \in [d]} |\partial_j T_N(\cdot)|\|_{L^\infty}$$
$$= \max_{j \in [d]} \|\partial_j T_N(\cdot)\|_{L^\infty} \leq 4\pi N \|T_N(\cdot)\|_{L^\infty},$$

which ends the proof. $\qquad\square$

