# OpenReview forum: "Finite Sample Bounds for Non-Parametric Regression: Optimal Sample Efficiency and Space Complexity"
_TMLR — Accepted by TMLR_

### Review · Reviewer_UQLL · 2026-04-23

**Summary Of Contributions:**

This paper proposes the first parametric algorithm that achieves minimax-optimal sample complexity for nonparametric regression under sub-Gaussian noise and passive (non-adaptive) design. The authors provide a fully finite-sample analysis, including high-probability, second-order (Bernstein-type) error bounds that make the bias-variance trade-off explicit. Compared with traditional non-parametric methods, they use a lightweight predictor whose memory and computational costs at prediction time depend solely on the number of parameters. The authors prove this complexity is information-theoretically optimal and validate their theoretical findings empirically, demonstrating competitive error rates with significantly reduced computational overhead.

**Additional Comments:**

On a more conceptual level, I have a high-level question regarding the framing. Since it appears we can effectively transform the non-parametric problem into a linear regression framework (for example, by using Fourier series), I find myself wondering: what is the fundamental advantage of retaining the non-parametric regression framing?

**Audience:**

Yes

**Audience Explanation:**

Non-parametric regression is a highly important area of machine learning. Using a parametric method to solve it while achieving minimax-optimal sample complexity and optimal computational efficiency is fairly significant and will undoubtedly be of interest to the TMLR audience.

**Broader Impact Concerns:**

As this is primarily a theoretical and methodological contribution, it does not present any obvious ethical implications or direct negative societal impacts.

**Claims And Evidence:**

Yes

**Claims Explanation:**

I am not an expert in this area. But the authors provide comprehensive theoretical results supported by rigorous proofs. Additionally, they include empirical experiments that effectively validate their claims.

**Requested Changes:**

1. It would be very helpful if you could expand the caption for Figure 1 to provide a bit more detail. Making the figure more self-contained would greatly improve readability for the audience.

2. I noticed that the citations do not have hyperlinks on my end. This might just be an issue with my PDF viewer, but it would be great if you could make them active in the compiled document.

3. In the related work, you mention other parametric methods that operate in this setting but cannot approximate derivatives. As someone who isn't an expert in this specific sub-field, I would love a bit more intuition here. Could you elaborate on how important it is to approximate derivatives in practice?

4. I noticed that the current evaluations primarily focus on comparisons with non-parametric methods. It would be wonderful to see a brief discussion of the theoretical results of existing parametric baselines as well. If feasible, including a comparison against them in the experiments would further strengthen the paper.

5. The reported improvements in computation and memory (table 1) feel slightly incremental as currently presented. It might strengthen the paper to elaborate on the significance of these savings. For instance, how might these theoretical reductions alleviate tangible bottlenecks in real-world applications?

6. Conceptually, if this approach effectively resolves non-parametric regression using a parametric model, I find myself wondering: what is the ongoing advantage of retaining the non-parametric framing? It would be highly illuminating if you could add a brief discussion (perhaps in the introduction or conclusion) to help non-experts understand why studying this through the lens of non-parametric regression remains so valuable.

---

> ### Author Response · Authors · 2026-05-18
> **Authors' Response**
>
> - _"It would be very helpful if you could expand the caption for Figure 1 ... audience."_
>
> We expanded the caption of Figure 1 to make it self-contained. The revised caption now points to the formal definitions of the Dirichlet and de la Vallée-Poussin kernels, explains the role of the degree $N$, and summarizes the key qualitative difference between the two kernels: the oscillatory behavior of the Dirichlet kernel versus the better localization of the de la Vallée-Poussin kernel.
>
> - _"I noticed that the citations do not have hyperlinks on my end... document."_
>
> We thank the reviewer for pointing this out. We have updated the LaTeX compilation settings to ensure that citations are active hyperlinks in the revised PDF.
>
> - _"In the related work, you mention other parametric methods ... Could you elaborate on how important it is to approximate derivatives in practice?"_
>
> For Smoothing Splines and Local Polynomial Regression, key smoothing parameters typically depend on the specific order of the derivative being estimated, making it impossible to have a plug-in estimator that is simultaneously good for all derivatives (as pointed out by Liu and Li). The necessity of plug-in estimators is paramount in contemporary physics-informed machine learning applications. In these contexts, one often aims to solve a partial differential equation (PDE) using noisy point-wise samples distributed across a domain, including its boundaries. When the PDE involves complex boundary conditions—such as Neumann or Robin conditions—the governing equations depend not only on the primary function but also on its partial derivatives. Consequently, achieving high-fidelity approximations for both the function and its derivatives on the data-bearing surfaces is essential for ensuring the global consistency and accuracy of the PDE solution.
> We will add this reflection to the final version.
>
> - _"I noticed that the current evaluations primarily focus on comparisons with non-parametric methods. It would be wonderful to see a brief discussion of the theoretical results of existing parametric baselines as well. If feasible, including a comparison against them in the experiments would further strengthen the paper."_
>
> We agree that this comparison is useful for positioning the contribution. In the revised related-work section, we expanded the discussion of existing parametric approaches. We did not include them in the main empirical comparison because, to the best of our knowledge, their available worst-case guarantees do not match the minimax-optimal $L^\infty$ rates for simultaneous derivative estimation considered in this paper. Thus, their computational profiles are similar in spirit, but they solve a statistically weaker problem. We now make this distinction explicit rather than presenting the comparison as purely empirical.
>
>
> - _"The reported improvements in computation and memory (table 1) feel slightly incremental as currently presented. It might strengthen the paper to elaborate on the significance of these savings. For instance, how might these theoretical reductions alleviate tangible bottlenecks in real-world applications?"_
>
> We added a paragraph titled “Different notions of computational complexity” in the comparison with LPE. The new discussion emphasizes that prediction-time complexity is often the bottleneck in real-time or repeatedly queried systems, even when training can be performed offline. We now explicitly discuss settings such as streaming, edge deployment, and high-frequency trading, where prediction latency and memory footprint are critical constraints. The revised text also clarifies that DUPA’s prediction-time storage depends only on the number of learned parameters, rather than on the full sample size.
>
> - _"Conceptually, if this approach effectively resolves non-parametric regression using a parametric model, I find myself wondering: what is the ongoing advantage of retaining the non-parametric framing? It would be highly illuminating if you could add a brief discussion (perhaps in the introduction or conclusion) to help non-experts understand why studying this through the lens of non-parametric regression remains so valuable."_
>
> We clarified that “non-parametric” refers to the statistical problem class, not to the implementation of the estimator. The target function is not assumed to belong to a fixed finite-dimensional model; it is only assumed to lie in a smoothness class. The number of parameters in our estimator grows with the sample size/target accuracy, which is precisely the non-parametric regime. Therefore, the non-parametric framing remains essential for formulating minimax rates, lower bounds, and the bias-variance trade-off. The main message of the paper is that a carefully designed growing parametric representation can solve a non-parametric problem with optimal statistical rates while retaining the computational advantages of parametric prediction.

---

### Review · Reviewer_7Byq · 2026-04-25

**Summary Of Contributions:**

This paper proposes a Fourier-feature estimator for smooth nonparametric regression under uniform error. It presents minimax-optimal high-probability sup-norm rates for estimating a smooth function and its derivatives. It also proposes a convolution/projection trick using the de la Vallée-Poussin kernel to avoid misspecification in linear regression. It derives finite-sample sub-gaussian and Bernstein-type second-order bounds and memory/prediction-time complexity improvements over classical nonparametric estimators.

**Audience:**

Yes

**Audience Explanation:**

This paper is most relevant to researchers in statistics, approximation theory, learning theory, and continuous bandits/RL.

**Claims And Evidence:**

Yes

**Claims Explanation:**

The theoretical claims are well supported by coherent proof: It uses de la Vallée-Poussin approximation to make the regression target exactly linear in the Fourier basis and quasi-optimal design for variance control. The main minimax-rate and the dependence on problem-parameter looks correct.

The drawback is that the empirical evidence is weaker. The experiments use only a one-dimensional audio-signal regression task with synthetic Gaussian noise and compare against NW/LPE, while omitting kernel ridge regression due to computational cost. This can work as a sanity check, but not strong evidence for the broader claims about modern ML, RL, or high-dimensional settings.

**Requested Changes:**

I would recommend stronger experiments: higher-dimensional synthetic benchmarks, derivative-estimation evaluations, memory measurements, ablations comparing Dirichlet and de la Vallée-Poussin kernels, and sensitivity to N and the noise level. It would also be nice to present clearer algorithmic details for implementing the perturbation distributions and optimal design.

---

> ### Author Response · Authors · 2026-05-18
> **Authors' Response**
>
> - _"I would recommend stronger experiments: higher-dimensional synthetic benchmarks, derivative-estimation evaluations, memory measurements, ablations comparing Dirichlet and de la Vallée-Poussin kernels, and sensitivity to N and the noise level. It would also be nice to present clearer algorithmic details for implementing the perturbation distributions and optimal design."_
>
> We appreciate the reviewer's constructive suggestions to strengthen the empirical evaluation. Following their specific recommendations, we have expanded our empirical evaluation and added Appendix H in the revised manuscript. This new section includes:
>
> (1) Higher-dimensional synthetic benchmarks evaluating multivariate performance.
>
> (2) Derivative-estimation evaluations.
>
> (3) An ablation study comparing the performance of our algorithm against the Dirichlet kernel.
>
> The new experiments show that the DVP version consistently outperforms the Dirichlet variant in both $L^2$ and $L^\infty$ metrics, for both function and derivative estimation. This supports the theoretical role of the DVP kernel in controlling uniform approximation error.
>
> This is supported by Table 2 in Appendix H, which reports both $L^2$ and $L^\infty$ errors for $f$ and $\partial f/\partial x_1$ over several sample sizes.
>
> We believe these additions make the paper much more comprehensive.

---

### Review · Reviewer_LJfc · 2026-05-04

**Summary Of Contributions:**

Summary: The paper proposes DUPA (Derivative-Uniform Parametric Approximation), a Fourier-based parametric estimator for smooth nonparametric regression with a periodic function under passive design. The DUPA method uses projection by convolution with the de la Vallée-Poussin kernel to transform the target into a finite-dimensional trigonometric polynomial, then performs least squares with a quasi-optimal design. The paper presents high-probability finite-sample uniform error bounds for the function and its derivatives. Experiments compare DUPA with Nadaraya-Watson and local polynomial estimators.

Strength:
(1) The idea of using the de la Vallee-Poussin kernel for approximation that is close in the supremum norm is valid. The paper also explains why the ordinary  Fourier projection is insufficient for deriving uniform bound and hence motivates the use of the de la Vallee-poussin kernel.

(2) The uniform estimation of the function $f$ and its derivatives is stronger than standard mean-squared estimation.

Weakness/Concerns:
(1) The proof of Theorem 5 is bounding the difference (left-hand side of equation (16)) while only taking into account the original sub-Gaussian observation noise (equation (13)). But the random perturbation $f(x+\eta)$ also adds extra variance. This might be problematic.

(2) The main results assume periodic boundary conditions on $[-1,1]^d$ and a non-periodic extention requires querying on a larger domain $[-3,3]^d$. In practical settings, is it always possible to query outside the target domain?

(3) There are a few typos acorss the main text and appendix.

**Audience:**

Yes

**Audience Explanation:**

The topic of nonparametric regression is interesting for the audience of TMLR. The problem and proposed method is well-motivated in the paper.

**Claims And Evidence:**

No

**Claims Explanation:**

The proof of Theorem 5 (main contribution) needs some clarification.

**Requested Changes:**

The three points in "Weakness" should be addressed.

---

> ### Author Response · Authors · 2026-05-18
> **Author's response**
>
> - _"The proof of Theorem 5 is bounding the difference (left-hand side of equation (16)) while only taking into account the original sub-Gaussian observation noise (equation (13)). But the random perturbation
>  also adds extra variance. This might be problematic."_
>
> We thank the reviewer for this observation. It is correct that the random perturbation introduces additional variance, and we agree with the reviewer that the previous equation did not clearly address this point. Specifically, the resulting sub-Gaussian constant scales, roughly, as $\Lambda_1 (\sigma+1)$ rather than $\sigma$ alone (similar form for the variance). For big values $\sigma\ge 1$ (resp. $\gamma\ge 1$), the regime we meant to consider initially, the samples complexity magnitude does not change, as $\Lambda_1 (\sigma+1)\le 2\Lambda_1\sigma$ and $\Lambda_1$ is a small constant. On the other side, the result is not valid for $\sigma \to 0$: we have clarified this point in all the theorems of the version just uploaded. While this regime is restrictive, we emphasize that classical work on nonparametric regression usually take $\gamma=1$.
>
> - _"The main results assume periodic boundary conditions on $[-1,1]^d$
>  and a non-periodic extention requires querying on a larger domain $[-3,3]^d$
> . In practical settings, is it always possible to query outside the target domain?"_
>
> The reviewer raises an important practical point regarding domain boundaries. While querying outside the target domain may not always be feasible in practice, we can bypass this restriction entirely by using extension methods that operate strictly on interior data. Specifically, Hestenes-type extension methods guarantee that a smooth function $f$ on a compact domain (e.g., $[-1,1]^d$) can be extended to a larger domain while preserving its smoothness class. Crucially, evaluating the extended function at an external point requires only a linear combination of queries \textit{inside} the original domain. For example, to extend a function $f$ from $[0,1]$ to $[-1,1]$, the extension for $x < 0$ is defined as:
> $$\tilde{f}(x) = \sum_{i=1}^{k+1} c_i f(-\lambda_i x)$$
> where $\lambda_i > 0$ ensures that $-\lambda_i x \in [0,1]$. Thus, no external queries are actually performed. We have added a discussion on Hestenes extensions in Appendix G of the revised manuscript to clarify this practical workaround.

---

### Decision · Action_Editor_L6q3 · 2026-06-15

**Recommendation:** Accept with minor revision

**Audience:**

Yes

**Audience Explanation:**

This paper considers the classical nonparametric regression problem under the supremum norm, and finds that the de la Vallee-Poussin kernel is more suitable than the ordinary Fourier projection for approximation. Both the problem and the findings are interesting to the TCS and statistics community.

**Claims And Evidence:**

Yes

**Claims Explanation:**

The theoretical claims of this paper are supported by rigorous proofs checked by the referees. Numerical experiments, including extensions and ablation studies added during the rebuttal, also support the theoretical claims and demonstrate the usefulness of the proposed approach. Some additional clarification on the linear-combination step, as suggested by one referee, will be helpful in the minor revision.